# *PITAR*, a DNA damage-inducible cancer/testis long noncoding RNA, inactivates p53 by binding and stabilizing *TRIM28* mRNA

**Samarjit Jana[1], Mainak Mondal[1], Sagar Mahale[2], Bhavana Gupta[1], Kaval Reddy Prasasvi[1], Lekha Kandasami[1], Neha Jha[1], Abhishek Chowdhury[1], Vani Santosh[3], Chandrasekhar Kanduri[2], Kumaravel Somasundaram[1]\***

[1]Department of Microbiology and Cell Biology, Indian Institute of Science Bangalore, Bangalore, India; [2]Department of Medical Biochemistry and Cell Biology, Institute of Biomedicine, University of Gothenburg, Gothenburg, Sweden; [3]Department of Neuropathology, National Institute of Mental Health and Neurosciences, Bangalore, India

**\*For correspondence:**
skumar1@iisc.ac.in

**Abstract** In tumors with WT p53, alternate mechanisms of p53 inactivation are reported. Here, we have identified a long noncoding RNA, *PITAR* (*p53 Inactivating TRIM28 Associated RNA*), as an inhibitor of p53. *PITAR* is an oncogenic Cancer/testis lncRNA and is highly expressed in glioblastoma (GBM) and glioma stem-like cells (GSC). We establish that *TRIM28* mRNA, which encodes a p53-specific E3 ubiquitin ligase, is a direct target of *PITAR*. *PITAR* interaction with *TRIM28* RNA stabilized *TRIM28* mRNA, which resulted in increased TRIM28 protein levels and reduced p53 steady-state levels due to enhanced p53 ubiquitination. DNA damage activated *PITAR*, in addition to p53, in a p53-independent manner, thus creating an incoherent feedforward loop to inhibit the DNA damage response by p53. While *PITAR* silencing inhibited the growth of WT p53 containing GSCs in vitro and reduced glioma tumor growth in vivo, its overexpression enhanced the tumor growth in a *TRIM28*-dependent manner and promoted resistance to Temozolomide. Thus, we establish an alternate way of p53 inactivation by *PITAR*, which maintains low p53 levels in normal cells and attenuates the DNA damage response by p53. Finally, we propose *PITAR* as a potential GBM therapeutic target.

## eLife assessment

This **important** study reports, with **convincing** evidence, that a long non-coding RNA disrupts the activity of the tumor suppressor p53 to contribute to the growth and therapeutic response of glioblastoma. The work will be relevant to scientists working on non-coding RNAs and brain tumors.

## Introduction

Much of the human genome, once considered as 'junk DNA', is now shown to be pervasively transcribed into thousands of long noncoding RNAs (lncRNAs). Accumulated evidence over the last two decades implicated lncRNA in fundamental biological processes that play an essential role in several pathological conditions like cancer (*Djebali et al., 2012*; *Slack and Chinnaiyan, 2019*). LncRNA constitutes a major subset among ncRNAs and is arbitrarily defined as transcripts of longer than 500 bps, which are spliced, 5' capped, and poorly conserved across species (*Cabili et al., 2011*; *Derrien et al., 2012*; *Iyer et al., 2015*; *Mattick et al., 2023*). LncRNAs function through multiple mechanisms,

such as their specific interactions with DNA, RNA, and proteins, that can modulate chromatin structure, regulate the assembly and function of membrane-less bodies, alter the stability and translation of mRNAs and interfere with their signaling pathways (*Statello et al., 2021*).

The tumor suppressor protein p53 plays an important role in preserving genome integrity and inhibiting malignant transformation (*Levine, 1997*). p53, which gets activated in response to stress like DNA damage, activates the transcription of many protein-coding genes that control several cellular processes like cell cycle, programmed cell death, and senescence (*Riley et al., 2008*; *Beckerman and Prives, 2010*; *Bieging and Attardi, 2012*). p53 is mutated in more than 50% of human cancers (*Vogelstein et al., 2000*; *Vousden and Lane, 2007*). In cancers that do not carry mutations in p53, the inactivation occurs through other genetic or epigenetic alterations (*Olivier et al., 2002*; *Vousden and Lu, 2002*; *Mitra et al., 2021*).

LncRNAs have been shown to play an important role in the p53 network. While several lncRNAs such as *LincRNA-p21*, *DINO*, *PANDA*, *LINC-PANT*, *GUARDIN*, *NEAT1*, *NBAT1* (*Mitra et al., 2021*), and *PVT1* are shown to function as downstream effectors of p53, other lncRNA like *MEG3*, *MALAT1*, *H19*, *Linc-ROR*, and *PSTAR* act as upstream regulators of p53 (*Jain, 2020*). Our study identified a lncRNA called *p53 Inactivating TRIM28 Associated RNA* (*PITAR*) as a p53 inactivator with a protumorigenic function. *PITAR* is highly expressed in glioblastoma (GBM) and glioma stem-like cells (GSCs) and interacts with *TRIM28* mRNA, which encodes a p53-specific E3 ubiquitin ligase. TRIM28 inhibits p53 through HDAC1-mediated deacetylation and direct ubiquitination in an Mdm2-dependent manner (*Wang et al., 2005*). *PITAR-TRIM28* interaction stabilized *TRIM28* mRNA, resulting in higher levels of TRIM28 protein that promoted ubiquitin-mediated degradation of p53. We also found that *PITAR* is essential for glioma tumor growth, and *PITAR* is induced by DNA damage in a p53-independent manner. Thus, our study discovered *PITAR* as an inhibitor of p53 via a unique mechanism of interaction with *TRIM28* mRNA and a potential target for developing novel therapy.

## Results

### Identification of *FAM95B1/PITAR*, a conserved cancer/testis lncRNA that promotes cell proliferation in GBM

In a recent study involving our group's transcriptional profiling of mRNAs and lncRNAs, we identified several GBM-specific clinically relevant lncRNA regulatory networks (*Paul et al., 2018*). To identify GBM-associated lncRNAs with functional relevance to glioma-stem-like cells (GSCs) biology, we integrated the differentially regulated RNAs (DEGs) from GBM vs control brain samples (TCGA; *Supplementary file 1*) with DEGs from GSC vs differentiated glioma cells (DGCs; *Suvà et al., 2014*; *Figure 1A*). The GBM and GSC integrated analysis revealed three interesting lncRNA *PVT1*, *H19*, and *FAM95B1*, showing significant upregulation in both GBM and GSCs (*Figure 1A*). Of the three GBM and GSC-specific regulated lncRNAs, *PVT1* and *H19* have been extensively investigated for their role in cancer development and progression (*Xue et al., 2018*; *Azab and Azzam, 2021*; *Li et al., 2022*; *Lv et al., 2022*; *Wang et al., 2022*), whereas *FAM95B1* (ENSG00000223839.7, NONHSAG052279.2, Lnc-ANKRD20A3-51) has not been explored for its role in cancer. We have named the *FAM95B1* lncRNA as *PITAR* as we found it has a strong functional association with the well-known tumor suppressor gene p53.

*PITAR* is significantly upregulated in multiple GBM cohorts (*Figure 1B–E*) and patient-derived GSCs (*Figure 1F and G*). The *PITAR* transcript also showed a one-to-one significant negative correlation between GSCs and DGCs (*Figure 1—figure supplement 1A*). *PITAR* promoter harbored active histone marks (H3k27Ac) in GSCs but not in DGCs (*Figure 1—figure supplement 1B*); thus, H3K27ac levels at the *PITAR* promoter in GSCs correlate with its expression status. Additional independent validation confirmed its higher expression in multiple glioma cell lines compared to immortalized astrocytes (*Figure 1H*) and in multiple patient-derived GSC lines (*Figure 1I*). Pan cancer data analysis revealed specific upregulation of *PITAR* in GBM and low-grade glioma (LGG) except in neuroendocrine tumor Pheochromocytoma and Paraganglioma (PCPG) (*Figure 1J*). Further, the interrogation of the Genotype-Tissue Expression (GTEx) of normal tissues revealed a highly tissue-restricted expression of *PITAR* in the testis, thus resembling the expression properties of a cancer-testis antigen (*Figure 1K*).

*PITAR*, located in human chromosome 9, has a 4236-nucleotide-long transcript (Transcript: ENST00000455995.5) with four exons (*Figure 2—figure supplement 1A*). The noncoding nature of

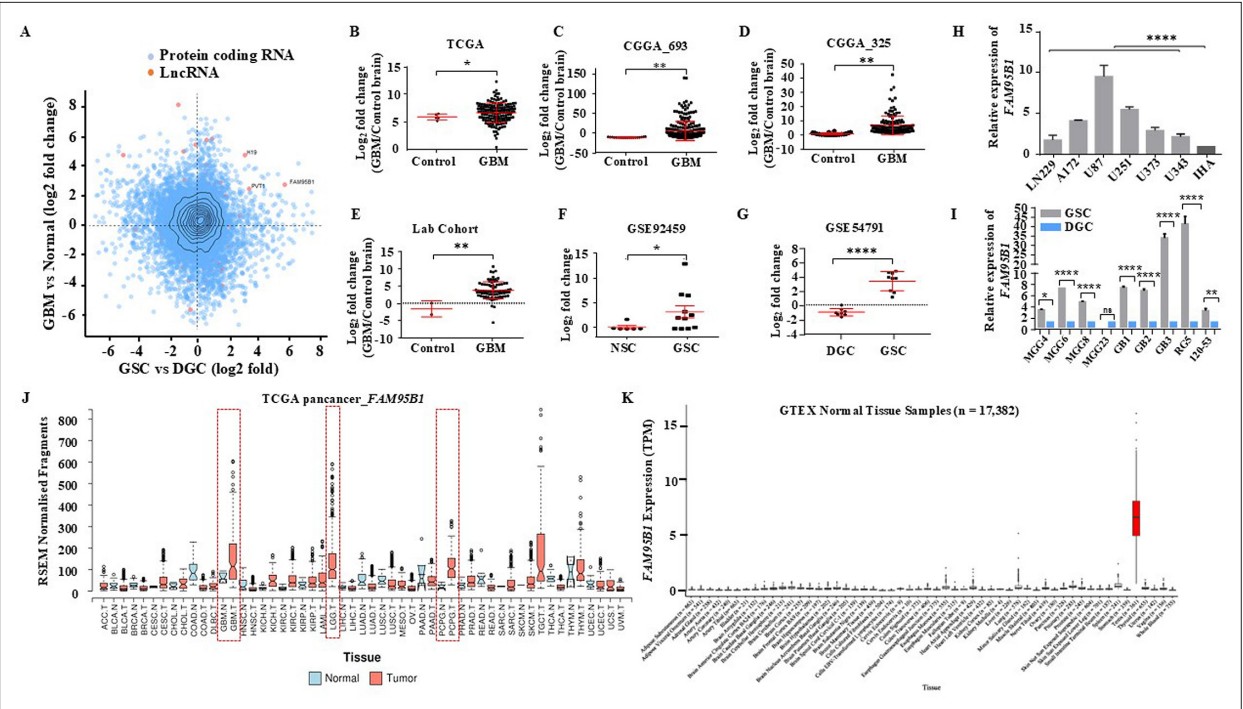

**Figure 1.** Identifying Glioblastoma stemcell-specific and conserved cancer/testis LncRNA. (**A**) The Scatterplot depicts the differentially regulated protein-coding (Blue dots) and lncRNA (Red dots) transcripts. The X-axis shows a differentially regulated gene (log2 fold change) in GBM vs. normal (TCGA patient cohort), and the Y-axis represents differentially regulated genes (log2 fold change) in the GSC Vs. DGC dataset (GSE54791). (**B–D**) The expression in log2 fold change of *FAM95B1* (*PITAR*) was shown in TCGA and two patient cohorts of the CGGA dataset (CGGA_693 & CGGA_325). GlioVis was used to obtain the gene expression matrix, and a t-test was performed using GraphPad Prism v6. (**E**) Plot depicts log2 fold change of *FAM95B1* (*PITAR*) in our patient cohort (normal, n=3 and GBM, n=79). Data are shown as mean ± SD and an unpaired t-test was performed using GraphPad Prism v6 (**p-value <0.01). (**F**) Expression of *PITAR* in GSC vs. NSC dataset (GSE92459). (**G**) Expression of *FAM95B1* (*PITAR*) in GSC vs. DGC dataset (GSE54791). Data are shown as mean ± SD and a unpaired t-test was performed using GraphPad Prism v6 (*p-value <0.05, ****p-value <0.0001). (**H**) Relative expression of *FAM95B1* (*PITAR*) was quantified by qRT-PCR in different Glioblastoma cell lines and immortalized human astrocytes (IHA). Data are shown as mean ± SD (n=3) and statistically analyzed with one-way ANOVA (****p-value <0.0001). (**I**) Relative expression of *FAM95B1* (*PITAR*) was measured in seven GSCs and their corresponding DGCs using the qRT-PCR method. Data are shown as mean ± SD (n=3) and an unpaired t-test was performed using GraphPad Prism v6 (*p-value <0.05, ****p-value <0.0001). (**J**) Relative expression (RPKM) of *FAM95B1* (*PITAR*) across different cancer types in the TCGA Pan-cancer cohort. (**K**) Expression in TPM of *FAM95B1* (*PITAR*) amongst the GTEx normal bulk tissue RNA-seq dataset. Data are shown as mean ± SD and a unpaired t-test was performed using GraphPad Prism v6 (*p-value <0.05, **p-value <0.01, ****p-value <0.0001).

The online version of this article includes the following figure supplement(s) for figure 1:

**Figure supplement 1.** Glioblastoma stem cell-specific expression of FAM95B1 (PITAR).

*PITAR* was confirmed by the Coding Potential Assessing Tool (CPAT) (https://www.ncbi.nlm.nih.gov/orffinder/; data not shown). The subcellular localization by fractionation followed by RT-qPCR identified that *PITAR* is primarily located in the nucleus (***Figure 2A***). Similarly, the RNA In situ hybridization using RNAScope technology revealed that *PITAR* is located 80% in the nucleus (***Figure 2B***). *PITAR* silencing by two different siRNAs, # 1 and # 2 (***Figure 2C***), reduced cell proliferation, viable cell count, colony formation, arrested cells in the G0/G1 phase, and induced apoptosis in U87 glioma cell line (***Figure 2D, E, F, G and H***). *PITAR* silencing in U343 glioma cells also showed similar results (***Figure 2—figure supplement 1B,C,D,E and F***). Further, *PITAR* silencing showed more sensitivity to DNA-damaging agents, Adriamycin, and Temozolomide (***Figure 2I***). Interestingly, *PITAR* silencing did not affect the viable cell count in immortalized astrocytes (***Figure 2J***). Overall, *PITAR* is a cancer-testis lncRNA that displays GBM and GSC-specific higher expression. We also demonstrate that *PITAR* promotes the growth of glioma cells and confers resistance to DNA-damaging agents.

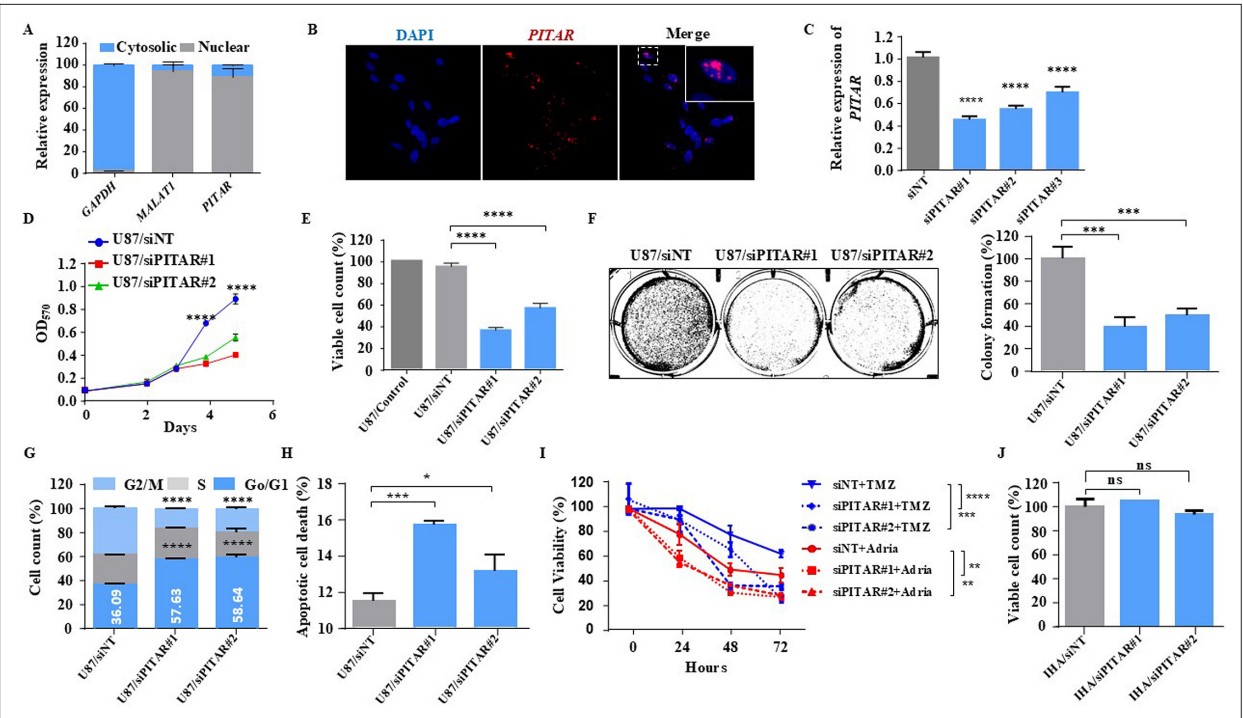

**Figure 2.** Glioblastoma cell proliferation and chemosensitivity altered by *PITAR* silencing. (**A**) Plot depicts subcellular fractionation of U87 cells followed by quantification using qRT-PCR. MALAT1 is a positive control for nuclear gene expression, and GAPDH is a positive control for cytoplasmic gene expression. Data are shown as mean ± SD (n=3) and an unpaired t-test was performed using GraphPad Prism v6 (*p-value <0.05, **p-value <0.01, ****p-value <0.0001). (**B**) The RNAScope images of *PITAR* (red) and the DAPI (nucleus, blue) counterstain in U87 cells. The indicative scale bar on the images is 50 µm. (**C**) The knockdown efficiency of siRNAs (siPITAR#1, siPITAR#2, and siPITAR#3) against *PITAR* was measured by qRT-PCR. Data are shown as mean ± SD (n=3) and an unpaired t-test was performed using GraphPad Prism v6 (****p-value <0.0001). (**D**) The cell proliferation was measured by MTT assay upon *PITAR* knockdown in U87 cells. (**E**) The viable cell count was measured using a viable cell counter following the Trypan blue method. Data are shown as mean ± SD (n=3) and an unpaired t-test was performed using GraphPad Prism v6 (****p-value <0.0001). (**F**) Colony formation assay was performed upon *PITAR* knockdown compared to siNT in U87 cells. Data are shown as mean ± SD (n=3) and a unpaired t-test was performed using GraphPad Prism v6 (***p-value <0.001). (**G**) Cell cycle analysis was performed in *PITAR*-silenced U87 cells. Data are shown as mean ± SD (n=3) and an unpaired t-test was performed using GraphPad Prism v6 (****p-value <0.0001). (**H**) The apoptotic cell death was measured by Annexin V/PI staining in *PITAR*-silenced U87 cells. Data are shown as mean ± SD (n=3) and an unpaired t-test was performed using GraphPad Prism v6 (*p-value <0.05, ***p-value <0.001). (**I**) The chemosensitivity upon *PITAR* silencing was measured by MTT assay against Adriamycin (0.25 µg/ml) and Temozolomide (300 µM) in U87 cells compared to control cells. Data are shown as mean ± SD (n=3) and an unpaired t-test was performed using GraphPad Prism v6 (**p-value <0.01, ***p-value <0.001, ****p-value <0.0001). (**J**) The viable cell count of human astrocytes (IHA) was measured upon *PITAR* silencing. Data are shown as mean ± SD (n=3) and "ns" represents not significant.

The online version of this article includes the following figure supplement(s) for figure 2:

**Figure supplement 1.** Silencing of PITAR inhibits cell growth in U343 glioma cells.

## *TRIM28* (tripartite motif containing 28) mRNA is the direct target of *PITAR*

To identify *PITAR* interacting RNAs, we carried out an integrated analysis of RNA-Seq data from Chromatin Isolation by RNA Purification (ChIRP), performed using biotinylated *PITAR*-specific antisense probes and RNA-Seq data of *PITAR* silenced glioma cells. ChIRP was carried out using an odd (n=7) and even (n=7) set of biotinylated *PITAR*-specific antisense probes (***Figure 2—figure supplement 1A***). RNA-Seq data of ChIRP RNA and further independent validation of ChIRP RNA with RT-qPCR showed an efficient pulldown of *PITAR* by both odd and even probes compared to LacZ (Control) probes (***Figure 3A and B***). The ChIRP data demonstrated that 827 mRNAs were enriched in the pulldowns using even and odd antisense probes compared to LacZ probes (***Supplementary file 2***). To choose the physiologically relevant target(s) for further studies, we intersected the ChIRP RNAs with (i) GBM-associated differentially expressed transcripts that show a significant positive correlation (p<0.05 and *r*>0.25) with *PITAR* transcript (***Supplementary file 3***), (ii) GBM upregulated transcripts

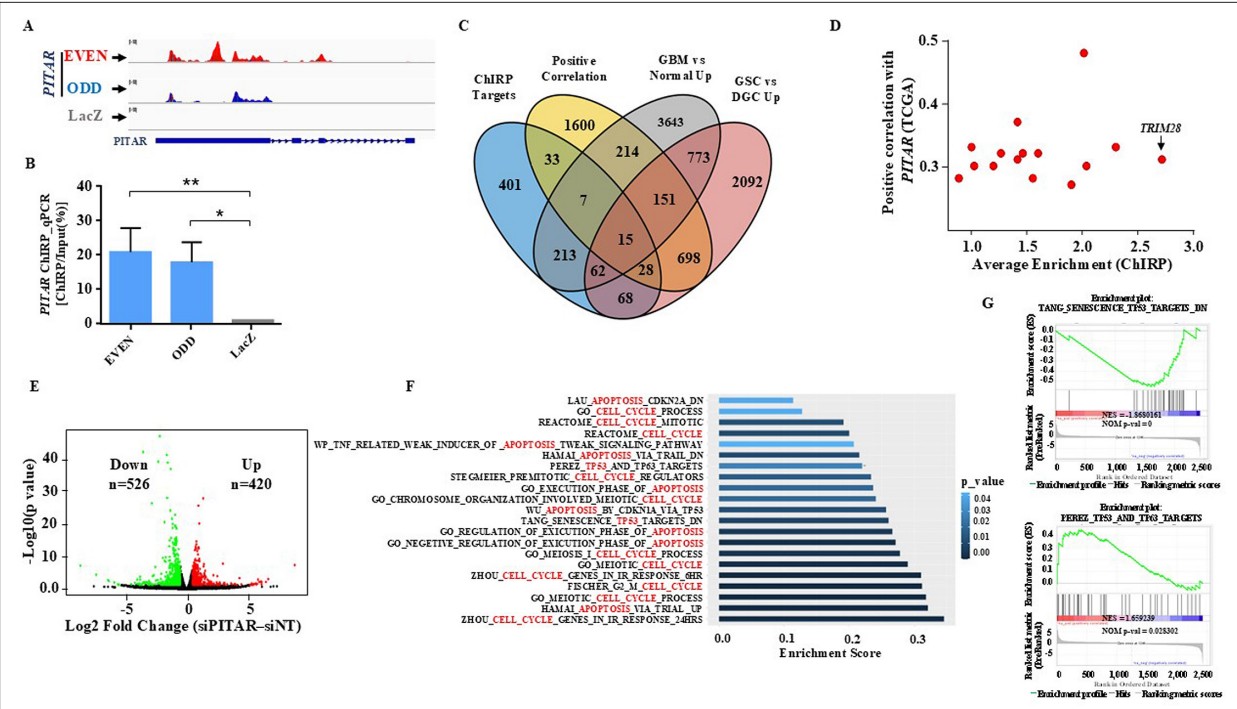

**Figure 3.** Identification of *PITAR* targets. (**A**) Genomic track for *PITAR* derived from ChIRP-RNA sequencing using Odd, Even, and LacZ antisense probe. (**B**) *PITAR* Pulldown by ChIRP assay was quantified using qRT-PCR. (**C**) The Venn diagram represents the association of four datasets (ChIRP enriched genes, *PITAR* positive correlated genes from TCGA, GBM vs. normal upregulated genes from TCGA, and GSC vs. DGC upregulated genes from GSE54791). (**D**) The selected 15 genes from the Venn diagram are plotted in the scatter plot, and an arrow marked *TRIM28* as a selected target. (**E**) The volcano plot depicts up-regulated (n=420) and down-regulated (n=526) genes upon *PITAR* knockdown compared to siNT. The gene expression matrix between siPITAR and siNT was used to construct a volcano plot to visualize differentially expressed genes. (**F**) Gene set enrichment analysis (GSEA) of differentially regulated genes was performed based on *PITAR* expression level at log2fold >0.58 and p<0.05. (**G**) The GSEA plots depict the enrichment of p53 up and down target gene sets, results derived from *PITAR*-silenced U87 cells. Data are shown as mean ± SD (n=3). ***p-value <0.001, **p-value <0.01, *p-value <0.05.

(*Supplementary file 4*), and (iii) GSC upregulated transcripts (*Supplementary file 5*). This analysis identified 15 transcripts as potential targets of *PITAR* (*Figure 3C and D*). Parallely, RNA-Seq data of *PITAR* silenced U87 cells identified 946 differentially regulated genes (526 upregulated and 420 down-regulated; *Figure 3E*; *Supplementary file 6*). Gene ontology analysis of DEGs showed significant enrichment of biological processes such as 'cell cycle,' 'apoptosis,' and 'p53' (*Figure 3F*). Furthermore, the Gene Set Enrichment Analysis (GSEA) showed significant enrichment of p53-regulated gene networks (*Supplementary file 7*; *Figure 3G*), indicating that *PITAR* executes its functions by interfering with p53 functions.

To explore the functional role of *PITAR* in p53-dependent gene expression, we chose *TRIM28* among the ChIRP targets considering its role as p53-specific E3 ubiquitin ligase (*Wang et al., 2005*). TRIM28, also known as KAP1 (Krüppel-Associated Box (KRAB)-Associated Protein 1), is a multidomain protein involved in various biological functions (*Czerwińska et al., 2017a*). TRIM28 inhibits p53 through HDAC1-mediated deacetylation and direct ubiquitination in an MDM2-dependent manner (*Wang et al., 2005*). To begin with, we assessed the expression of the TRIM28 transcript in glioma. *TRIM28* is significantly upregulated in multiple GBM cohorts (*Figure 4—figure supplement 1A,B,C,D and E*). Consistent with the ChIRP data, *TRIM28* and *PITAR* transcripts showed a significant positive correlation in all GBM cohorts (*Figure 4—figure supplement 1F,G,H,I and J*). Like *PITAR*, the *TRIM28* transcript is also expressed at higher levels in GSCs (*Figure 4—figure supplement 1K*), with a significant positive correlation between their expression in GSCs (*Figure 4—figure supplement 1L*). Further, TRIM28 protein showed higher levels in GBM as measured by quantitative proteomics data from the Clinical Proteomic Tumor Analysis Consortium (CPTAC) and immunohistochemistry from the Protein atlas (*Figure 4—figure supplement 1M and N*).

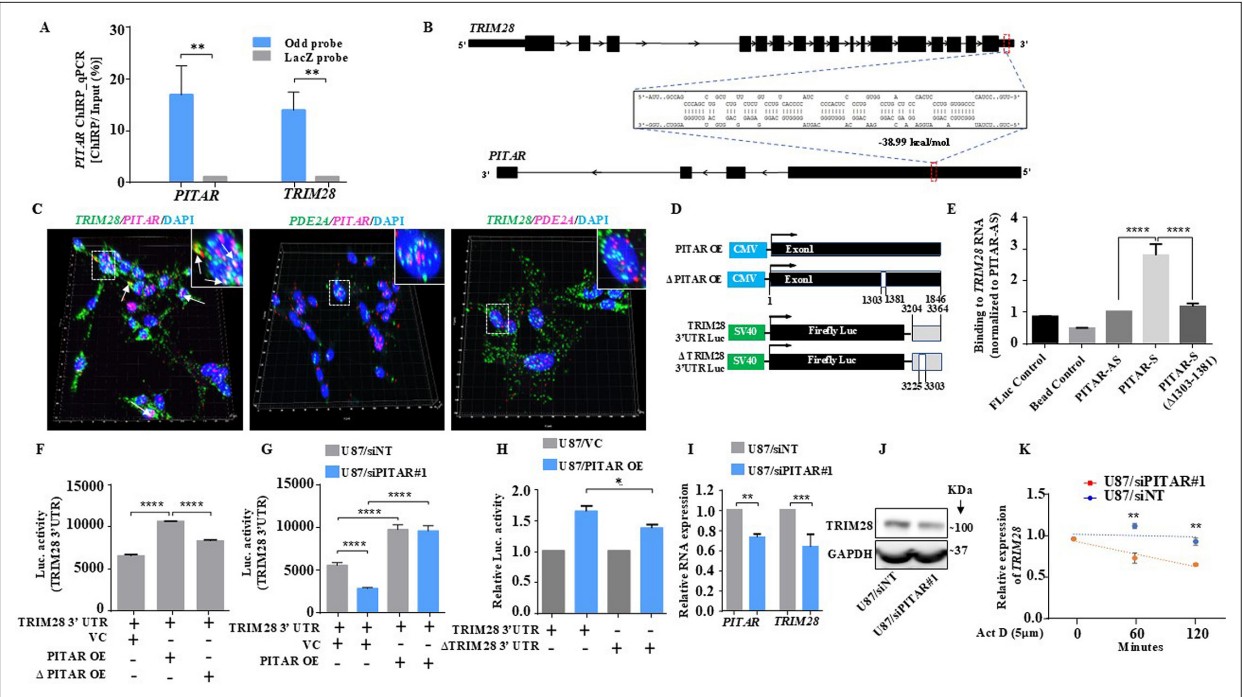

**Figure 4.** *PITAR* regulates the expression of *TRIM28* by physical interaction with *TRIM28*. (**A**) The qRT-PCR of *PITAR* and *TRIM28* RNA was performed in ChIRP-RNA pull-down samples. The Probes for Odd and LacZ were used to pull down endogenous *PITAR* and interacting *TRIM28* mRNA in U87 cells. (**B**) Schematic represents the predicted RNA–RNA interaction between *PITAR* and the 3'UTR of *TRIM28*. (**C**) RNAScope images of co-localized signals of *PITAR* (red) and *TRIM28* (green) in U87 cells. The panel shows the 3D reconstructed cell images of the merged 2D image (Imaris image analysis software). Yellow dots shown by white arrows depict co-localized *PITAR* (red) and *TRIM28* (green). Magnified co-localized puncta were shown in the inset at the upper right corners, indicated by a white dotted box. The panel shows the RNAScope images of the localization of *PITAR* (red), *TRIM28* (green), and *PDE2A* (green/red). *PDE2A* RNA was used as a negative control. The indicative scale bar on the images is 50 µm. (**D**) Schematic of vector plasmid construct for PITAR OE, ΔPITAR OE, TRIM28 3'UTR, and ΔTRIM28 3'UTR. (**E**) *PITAR* interaction with the *TRIM28* 3'UTR was measured using an in vitro RNA–RNA interaction assay and compared to a panel of control RNAs (PITAR antisense, Fluc control, Bead control). The binding affinity was quantified by qRT-PCR analysis of the *TRIM28*. Data were normalized to the PITAR-AS control. (**F**) The Luc activity of *TRIM28* 3'UTR was measured after the ectopic expression of *PITAR* and ΔPITAR in U87 cells using a luciferase reporter assay. (**G**) Luciferase assay was performed in *PITAR* silenced U87 cells co-transfected with VC and PITAR OE vector. (**H**) The Firefly luciferase activity was measured in U87 cells containing a deleted *PITAR* binding site of *TRIM28* 3'UTR (ΔTRIM28 3'UTR), co-transfected with VC and PITAR OE. (**I**) Relative expression of *TRIM28* in *PITAR*-silenced U87 cells was measured by qRT-PCR. (**J**) The TRIM28 protein expression was measured by immunoblotting. Data are shown as mean ± SD (n=3) and an unpaired t-test was performed using GraphPad Prism v6 (*p-value <0.05, **p-value <0.01, ****p-value <0.0001). (**K**) *TRIM28* transcript was measured at indicated time points post Actinomycin D (5 µg/ml) treatment in siNT and siPITAR-transfected U87 cells by qRT-PCR (n=3). The log2 ratio of the remaining *TRIM28* was plotted using linear regression after normalizing to the 0th hour of the respective condition using GraphPad Prism v6. Data are shown as mean ± SD (**p-value <0.01).

The online version of this article includes the following source data and figure supplement(s) for figure 4:

**Source data 1.** Raw unedited blots for *Figure 4J*.

**Source data 2.** Uncropped and labeled blots for *Figure 4J*.

**Figure supplement 1.** Clinical relevance of *PITAR* and *TRIM28* association.

**Figure supplement 2.** PITAR physically interacts with TRIM28 mRNA.

**Figure supplement 2—source data 1.** Raw unedited gels for *Figure 4—figure supplement 2B*.

**Figure supplement 2—source data 2.** Uncropped and labeled gels for *Figure 4-figure supplement 2B*.

Next, we investigated the specificity and impact of the interaction between *PITAR* and *TRIM28* transcripts. First, we show that the genomic track data displays the specific pulldown of *TRIM28* mRNA by *PITAR*-specific odd and even antisense probes compared to LacZ probes in the RNA-Seq data of the ChIRP assay (*Figure 4—figure supplement 2A*). Second, we also independently validate this finding by the efficient pulldown of *PITAR* by *PITAR*-specific probes compared to LacZ (Control) probes by RT-qPCR from the ChIRP RNA (*Figure 4A*). Noncoding RNAs interact with target mRNAs through direct or indirect RNA-RNA interactions (*Faghihi et al., 2008*; *Gong and Maquat, 2011*;

*Kretz et al., 2013*; *Engreitz et al., 2014*; *Mahale et al., 2022*). Enrichment of *TRIM28* mRNA in the *PITAR* ChIRP pull-down indicates the presence of potential *TRIM28* mRNA: *PITAR* lncRNA interactions. Hence, we checked for the possible direct interaction between *PITAR* and *TIRM28* by performing an RNA-RNA interaction analysis using the IntaRNA tool (http://rna.informatik.uni-freiburg.de/IntaRNA/Input.jsp, *Mann et al., 2017*; *Gawronski et al., 2018*). This analysis identified the most energetically favorable 80 bp interacting region between the 3' UTR of the *TRIM28* transcript and the first exon of *PITAR* (*Figure 4B*; *Supplementary file 8*). Next, we performed multiplexed RNAScope in U87 cells to evaluate the interaction between *PITAR* and *TRIM28* in the cellular environment. The co-staining of U87 cells for *TRIM28* transcript with the *PITAR* or *PDE2A* and *PITAR* with *PDE2A* demonstrated a specific colocalization of *PITAR* and *TRIM28* transcripts, mainly in the nucleus. The extent of colocalization was much greater than that expected from coincidental colocalization with a similar abundant transcript, such as *PDE2A* (*Figure 4C*; *Figure 4—figure supplement 2C*).

To further confirm the interaction between *PITAR* and *TRIM28*, we cloned the first exon under the CMV promoter (PITAR OE) for exogenous overexpression and in vitro synthesis of biotin-labeled sense *PITAR* RNA (*Figure 4D*). First, we checked the ability of biotin-labeled *PITAR* sense RNA (corresponding to the first exon made from the PITAR OE construct) to bring down the *TRIM28* mRNA from total RNA. *PITAR*-specific sense, but not antisense, biotinylated RNA brought down *TRIM28* RNA efficiently (*Figure 4E*, compare bar four with three). *PITAR* sense RNA with a deletion of the 80 bp interacting region failed to bring down *TRIM28* RNA (*Figure 4E*, compare bar five with four). The control biotin-labeled RNA (FLuc) and bead control did not bring down *TRIM28* mRNA as expected. Further, we performed an antisense oligo-blocking experiment to validate the *TRIM28* binding site on *PITAR*. Preincubation of fragmented total RNA with unlabeled *PITAR* antisense probe # 3, which is located close to the *TRIM28* binding region on the exon 1 of *PITAR*, inhibited the ability of *PITAR* biotinylated antisense (odd) probe set to bring down *TRIM28* (*Figure 4—figure supplement 2B*). Next, we tested the ability of *PITAR* to regulate luciferase activity from the TRIM28-3'UTR-Luc construct (*Figure 4D*). The exogenous overexpression of *PITAR* (PITAR OE), but not *PITAR* with a deletion of 1303–1381 nucleotides (ΔPITAR OE), significantly increased the luciferase activity from the TRIM28-3'UTR-Luc construct (*Figure 4F*, compare bars two and three with one). In contrast, *PITAR* silencing significantly decreased the luciferase activity from TRIM28-3'UTR-Luc but was rescued by the exogenous overexpression of *PITAR* (*Figure 4G*, compare bar two with one or four). In contrast, the ability of *PITAR* exogenous overexpression to increase the luciferase activity from the ΔTRIM28-3'UTR-Luc construct, which has a deletion of 80 bp region corresponding to the *PITAR* binding region, is significantly reduced (*Figure 4H*, compare bar four with two). Next, to study the impact of *PITAR* interaction on *TRIM28* expression, we measured *TRIM28* transcript and protein levels in *PITAR*-silenced conditions. *PITAR* silencing significantly reduced *TRIM28* mRNA (*Figure 4I*, compare bar four with three) and protein levels (*Figure 4I*, compare lane two with one). In actinomycin-treated U87 glioma cells, *PITAR* silencing reduced the *TRIM28* mRNA half-life significantly compared with control cells (*Figure 4J*, compare red line with blue line). From these results, we conclude that *PITAR* interaction stabilizes *TRIM28* mRNA to promote *TRIM28* expression.

### *PITAR* inhibits p53 protein levels by its association with *TRIM28* mRNA

*TRIM28* inhibits p53 through HDAC1-mediated deacetylation and direct ubiquitination in an Mdm2-dependent manner (*Wang et al., 2005*). We next investigated the impact of *PITAR* interaction with *TRIM28* on p53. *PITAR* silencing in U87 glioma cells increased the luciferase activity from PG13-Luc, a p53-dependent reporter (*el-Deiry et al., 1993*; *Figure 5—figure supplement 1A*), decreased *PITAR* and *TRIM28* mRNA levels, and increased *CDKN1A* mRNA levels with no change in *TP53* mRNA levels (*Figure 5A*). At the protein level, *PITAR* silencing decreased TRIM28 levels but increased p53 and p21 levels (*Figure 5B*, compare lanes two and three with one). *PITAR* silencing in U343 cells also showed similar results (*Figure 5—figure supplement 1B and C*). In contrast, exogenous *PITAR* overexpression increased *PITAR* and *TRIM28* mRNA levels and reduced *CDKN1A* mRNA levels without changing *TP53* mRNA levels (*Figure 5C*). Further, exogenous *PITAR* overexpression increased TRIM28 protein levels but decreased p53 and p21 protein levels (*Figure 5D*, compare lane two with one), thus indicating that *PITAR* reduces p53 protein levels by regulating *TRIM28*. *PITAR* overexpression in U343 cells also showed similar results (*Figure 5—figure supplement 1D and E*). In cycloheximide-treated U87 glioma cells, the half-life of p53 increased (1.20 hr) under *PITAR* silenced conditions compared

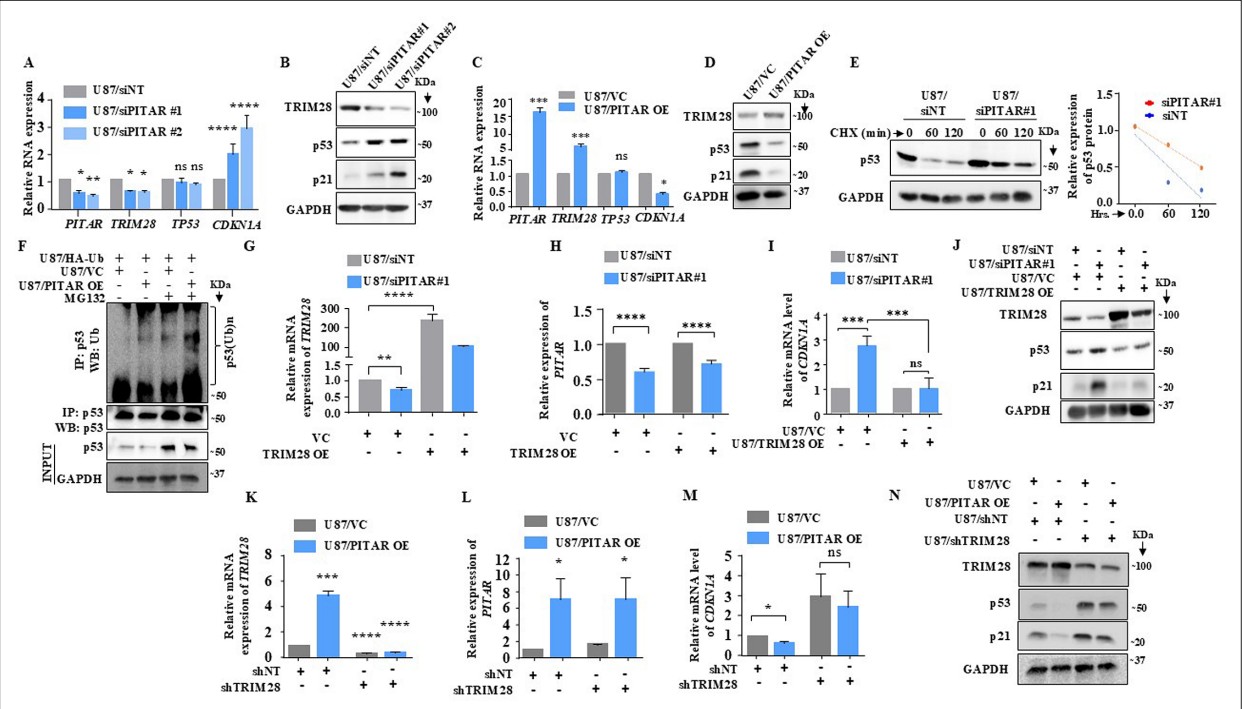

**Figure 5.** *PITAR* regulates wild-type p53 protein levels via *TRIM28*-mediated ubiquitination. (**A**) Relative expression of *PITAR*, *TRIM28*, *TP53*, and *CDKN1A* was quantified by qRT-PCR in *PITAR*-silenced U87 cells compared to siNT. (**B**) The protein expression of TRIM28, p53, and p21 was measured by immunoblotting in *PITAR*-silenced U87 cells compared to siNT. (**C, D**) Cells were transfected with pcDNA3.1-PITAR (PITAR OE)/ empty vector control plasmid (pcDNA3.1) and harvested 48 hr post-transfection for qRT-PCR (*PITAR*, *TRIM28*, *TP53*, and *CDKN1A*) and immunoblotting with indicated antibodies (TRIM28, p53, and p21). GAPDH served as the control. (**E**) The Half-life of the p53 protein was measured in *PITAR*-silencing (siPITAR) and control (siNT) U87 cells with the treatment of cycloheximide (CHX; 50 µg/mL). The relative expression of the remaining p53 was plotted using linear regression after normalizing to the 0th hour of the respective condition. (**F**) The endogenous level of p53 ubiquitination was measured in pcDNA3.1-PITAR (PITAR OE)/empty vector plasmid (pcDNA3.1) stable U87 cells by p53 immunoprecipitation followed by immunoblotting with the indicated antibodies in the presence and absence of MG132. (**G–I**) The relative expression of *PITAR*, *TRIM28*, and *CDKN1A* was measured by qRT-PCR in U87/siPITAR#1 and U87/siNT cells with exogenously overexpressed TRIM28 conditions. (**J**) The protein expression of TRIM28, p53, and p21 was measured by immunoblotting in U87/siPITAR#1 and U87/siNT cells with exogenously overexpressed TRIM28 condition. (**K–M**) The relative expression of *PITAR*, *TRIM28*, and *CDKN1A* was measured by qRT-PCR in U87/shTRIM28 and U87/shNT cells with exogenously overexpressed *PITAR* conditions. (**N**) The protein expression of TRIM28, p53, and p21 was measured by immunoblotting in U87/shTRIM28 and U87/shNT cells with exogenously overexpressed *PITAR* conditions. Data are shown as mean ± SD (n=3) and an unpaired t-test was performed using GraphPad Prism v6 (*p-value <0.05, **p-value <0.01,***p-value <0.001, ****p-value <0.0001).

The online version of this article includes the following source data and figure supplement(s) for figure 5:

**Source data 1.** Raw unedited blots for *Figure 5*.

**Source data 2.** Uncropped and labeled blots for *Figure 5*.

**Figure supplement 1.** PITAR inhibits wild-type p53 protein through TRIM28.

**Figure supplement 1—source data 1.** Raw unedited blots for *Figure 5—figure supplement 1*.

**Figure supplement 1—source data 2.** Uncropped and labeled blots for *Figure 5-figure supplement 1*.

**Figure supplement 2.** Inhibition of glioma cell growth by PITAR requires wild type p53.

**Figure supplement 2—source data 1.** Raw unedited blots for *Figure 5—figure supplement 2B*.

**Figure supplement 2—source data 2.** Uncropped and labeled blots for *Figure 5-figure supplement 2B*.

**Figure supplement 3.** Glioblastoma stem-like cell growth is induced by *PITAR* through p53 inactivation.

to control cells (0.50 hr; *Figure 5E*). In good correlation, *PITAR* overexpression increased the ubiquitinated p53 levels in MG132-treated U87 glioma cells compared to the control condition (*Figure 5F*, compare lane four with three).

To confirm that *TRIM28* mediates *PITAR* regulation of p53, we checked the ability of exogenously overexpressed *TRIM28* (using a 3'UTRless *TRIM28* construct) to rescue the phenotype in *PITAR*-silenced

cells. *TRIM28* overexpression resulted in a several-fold increase in *TRIM28* mRNA levels in U87/siNT cells (*Figure 5G*, compare bar three with one). While this increase was significantly affected by *PITAR* silencing, the *TRIM28* transcript levels remained high in U87/siPITAR cells (*Figure 5G*, compare bar four with two). As expected, *TRIM28* overexpression did not affect the *PITAR* mRNA levels in U87/siNT and U87/siPITAR cells (*Figure 5H*). In *TRIM28* overexpressing cells, *PITAR* silencing failed to increase luciferase activity from PG13-Luc (*Figure 5—figure supplement 1F*, compare bar four with three) and *CDKN1A* transcript levels (*Figure 5I*, compare bar four with three). More importantly, *PITAR* silencing failed to increase p53 and p21 protein levels in *TRIM28* overexpressing U87 glioma cells (*Figure 5J*, compare lane four with three). Next, we tested the ability of *PITAR* overexpression to inhibit p53 functions in *TRIM28* silenced cells. Exogenous overexpression of *PITAR* failed to increase the *TRIM28* in *TRIM28* silenced condition (*Figure 5K and L*). In addition, exogenous overexpression of *PITAR* failed to repress luciferase activity from PG13-Luc (*Figure 5—figure supplement 1G*, compare bar four with three), repress *CDKN1A* transcript levels (*Figure 5M*, compare lane four with three), and protein levels of p53 and p21 (*Figure 5N*, compare lane four with three) in TIRM28 silenced cells. These results establish that *PITAR* inhibits p53 through its interaction with *TRIM28* mRNA.

Next, we tested the requirement of WT p53 for the growth-promoting functions of *PITAR*, as shown above (*Figures 2 and 5*). *PITAR* silencing inhibited colony formation efficiently in U87/shNT cells but not in U87/shp53#1 (*Figure 5—figure supplement 2A,B and C*). As shown before, *PITAR* silencing significantly increased *CDKN1A* and *MDM2* transcript levels in U87/shNT cells but was compromised significantly in U87/shp53#1 (*Figure 5—figure supplement 2D*). We also used GSCs for this purpose. The ability of RG5, a patient-derived glioma stem cell line containing WT p53 (data not shown), to grow as a neurosphere was investigated in *PITAR*-silenced and overexpressed conditions. *PITAR* silencing inhibited the RG5 GSC growth in the neurosphere growth assay and limiting dilution assay (*Figure 5—figure supplement 3A,B,C and D*). *PITAR* silencing resulted in a decrease in *PITAR* and *TRIM28* but an increase in *CDKN1A* transcript levels (*Figure 5—figure supplement 3E*). As expected, there was no change in p53 transcript levels (*Figure 5—figure supplement 3E*). The exogenous over-expression of *PITAR* promoted the neurosphere growth by RG5 (*Figure 5—figure supplement 3F,G and H*) and increased *PITAR* and *TRIM28* transcript levels but decreased the *CDKN1A* transcript levels with no change in *TP53* transcript levels (*Figure 5—figure supplement 3I*). In contrast, the growth of MGG8, a patient-derived glioma stem cell culture containing mutant p53 (data not shown), is not affected by *PITAR* silencing (*Figure 5—figure supplement 3J,K,L,M and N*). We conclude from these results that *PITAR* growth-promoting functions of glioma cells require wild-type p53.

## *PITAR* is induced by DNA damage in a p53-independent manner, which in turn diminishes the DNA damage response by p53

As p53 is activated by DNA damage (*Shieh et al., 1997*), we investigated the ability of *PITAR* to inhibit DNA damage-induced p53. Adriamycin-induced luciferase activity from PG13-Luc (*Figure 6—figure supplement 1A*, compare bar four with three) and *CDKN1A* mRNA levels are significantly reduced by *PITAR* overexpression (*Figure 6A*, compare bar four with three). At the protein level, *PITAR* overexpression significantly reduced adriamycin-induced total p53, acetylated p53, and p21 levels (*Figure 6B*, compare lane four with three). In addition, the viable cell count increased significantly in adriamycin-treated cells upon *PITAR* overexpression (*Figure 6C*, compare bar four with three). Similarly, the extent of G2/M arrest seen in adriamycin-treated cells is reduced significantly (18.34 %) upon *PITAR* overexpression (*Figure 6D*; 45.80% is reduced to 37.40%).

To verify the possible existence of a negative feedback loop between *PITAR* and p53, we checked the ability of p53, expressed from a recombinant adenovirus, to induce *PITAR* transcript levels. We found that Ad-p53 induced *TP53* and *CDKN1A* mRNA and p53 protein levels compared to the control virus (Ad-GFP) but did not alter the *PITAR* and *TRIM28* transcript levels (*Figure 6E and F*), thus confirming that neither *PITAR* nor *TRIM28* is the direct target of p53. Interestingly, DNA damage by adriamycin treatment also induced *PITAR*, *TRIM28* transcript, and TRIM28 protein levels besides *CDKN1A* transcript and protein levels of p53 and p21 (*Figure 6G and H*). Our results also show that adriamycin treatment induced *PITAR* and *TRIM28* transcript levels efficiently in p53 silenced cells (*Figure 6I and J*, compare bar four with three), indicating DNA damage-mediated *PITAR* induction is p53 independent. Further, *PITAR* and *TRIM28* induction by DNA damage is dependent on ATM/ATR kinase as the pretreatment of glioma cells with CGK733, a small molecule inhibitor of ATM/

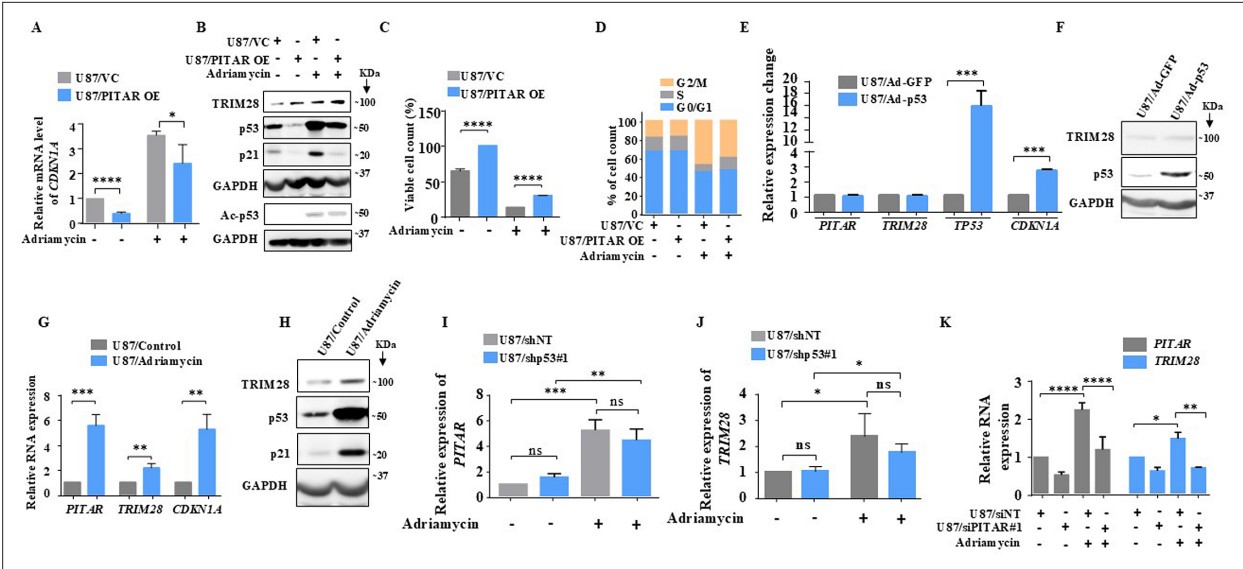

**Figure 6.** DNA damage-induced *PITAR* diminishes the DNA damage response by p53 through *TRIM28*. (**A**) The relative expression of *CDKN1A* was measured in the presence and absence of Adriamycin by qRT-PCR in PITAR OE/VC U87 cells. (**B**) The protein expression of TRIM28, p53, ac-p53, and p21 was measured in the presence and absence of Adriamycin by immunoblotting in PITAR OE/VC U87 cells. GAPDH served as the control. (**C**) The viable cell count was performed in the presence and absence of Adriamycin in PITAR OE/VC U87 cells. (**D**) The cell cycle analysis was performed in the presence and absence of Adriamycin in PITAR OE/VC U87 cells. (**E**) The relative expression of *PITAR*, *TRIM28*, *TP53*, and *CDKN1A* was measured by qRT-PCR in Ad-p53 and Ad-GFP-infected U87 cells. (**F**) The protein expression of TRIM28 and p53 was measured by immunoblotting in Ad-p53 and Ad-GFP-infected U87 cells. (**G**) The relative expression of *PITAR*, *TRIM28*, and *CDKN1A* was measured in the presence and absence of Adriamycin by qRT-PCR in U87 cells. (**H**) The immunoblot depicting the expression of TRIM28, p53, and p21 upon treatment of Adriamycin. (**I, J**) The qRT-PCR was performed to measure the relative expression of *PITAR* and *TRIM28* in the p53 knockdown condition upon Adriamycin treatment. (**K**) The relative expression of *PITAR* and *TRIM28* was measured by qRT-PCR in Adriamycin-treated *PITAR*-silenced U87 cells. Data are shown as mean ± SD (n=3) and an unpaired t-test was performed using GraphPad Prism v6 (*p-value <0.05, **p-value <0.01,***p-value <0.001, ****p-value <0.0001).

The online version of this article includes the following source data and figure supplement(s) for figure 6:

**Source data 1.** Raw unedited blots for *Figure 6*.

**Source data 2.** Uncropped and labeled blots for *Figure 6*.

**Figure supplement 1.** DNA damage-induced PITAR suppresses p53 respons to DNA damage.

**Figure supplement 1—source data 1.** Raw unedited blots for *Figure 6—figure supplement 1*.

**Figure supplement 1—source data 2.** Uncropped and labeled blots for *Figure 6-figure supplement 1*.

ATR kinase, prevented the adriamycin-mediated induction of *PITAR* and *TRIM28* transcript levels (*Figure 6—figure supplement 1B*) and TRIM28 protein level (*Figure 6—figure supplement 1C*) thus reiterating the importance of ATM/ATR kinases, as previously demonstrated (*Tibbetts et al., 1999*; *Cheng and Chen, 2010*), in DNA damage response pathway. Interestingly, the DNA damage-induced *TRIM28* transcript upregulation is found to be dependent on *PITAR* upregulation, as adriamycin failed to induce *TRIM28* in *PITAR*-silenced U87 cells (*Figure 6K*, compare bar eight with seven). This was further confirmed by the fact that adriamycin-induced luciferase activity from TRIM28 3'-UTR-Luc is inhibited by CGK733 treatment (*Figure 6—figure supplement 1D*, compare bar three with two), suggesting the requirement of *PITAR* for the *TRIM28* transcript upregulation in DNA-damaged cells. These results establish that *PITAR* is DNA damage-inducible in a p53-independent manner, and it inactivates p53 through its association with *TRIM28* mRNA.

The p53-Mdm2 autoregulatory negative feedback loop controls the extent and duration of p53 response upon DNA damage (*Wu et al., 1993*; *Haupt et al., 1997*; *Kubbutat et al., 1997*; *Zhang et al., 2009*). Since TRIM28 association with Mdm2 contributes to p53 inactivation (*Wang et al., 2005*; *Czerwińska et al., 2017a*), we hypothesized that *PITAR*, through its association with *TRIM28* may also contribute to the control of DNA damage response by p53. We tested this possibility by measuring the p53 response in *PITAR*-silenced cells. U87/siNT and U87/siPITAR#1 cells were exposed to 7 Gy of ionizing radiation, and steady-state levels of p53, Mdm2, and p21 proteins were determined at hourly

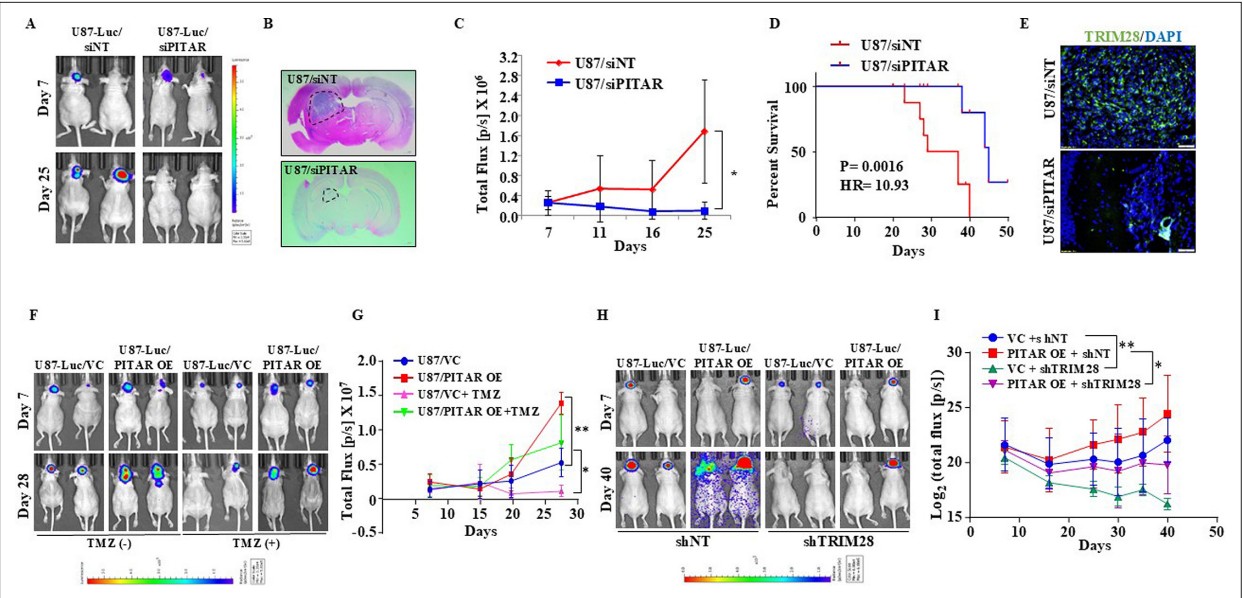

**Figure 7.** *PITAR* promotes glioma tumor growth and resistance to Temozolomide chemotherapy. (**A**) Mice (NIH nu/nu) were injected with siNT/ siPITAR#1 transfected U87-Luc cells (0.3x10⁶ cells/mice), and tumors were allowed to grow for 50 days (n=10), and the luminescence imaging was performed using the IVIS instrument (Perkin Elmer IVIS system). (**B**) H&E staining was performed in formalin-fixed tumor-bearing (siNT and siPITAR#1) mouse brain sections. (**C**) The tumor growth curve of siNT and siPITAR#1 tumor-bearing mice was quantified over time using IVIS. The difference between groups was statistically analyzed by ANOVA with Tukey's multiple comparisons test (*p<0.05).(**D**) The Kaplan–Meier graph shows the survival of mice-bearing tumors formed by siNT and siPITAR#1 cells (N=10/group), statistical differences were calculated with the Log-rank (Mantel-Cox) test (*p<0.05). (**E**) The mmunohistochemistry assay was performed to show the TRIM28 protein expression in the tumor tissue section derived from U87-Luc/ siNT and U87-Luc/siPITAR#1 tumors. The Green color represents the TRIM28 protein, and the blue depicts the nucleus stained with DAPI. Scale bar = 100 µm. (**F**) Mice (NIH nu/nu) were injected with U87-Luc/PITAR OE and U87-Luc/VC cells (0.3x10⁶ cells/mice, n=10), and tumors were allowed to grow for 30 days. The tumor-bearing mice were treated with 100 mg/kg TMZ in 25% DMSO saline solution after 11 days by intraperitoneal injection for one week, and the luminescence imaging was performed using the IVIS instrument. (**G**) The tumor growth curve of VC, PITAR OE, VC +TMZ, and PITAR OE +TMZ tumor-bearing mice was quantified over time using IVIS. The difference between groups was statistically analyzed by ANOVA with Tukey's multiple comparisons test (*p<0.05, ***p<0.001). (**H**) Mice (NIH nu/nu) were injected with U87-Luc/PITAR OE +shNT, U87-Luc/VC +shNT, U87-Luc/ PITAR OE +shTRIM28 and U87-Luc/VC +shTRIM28 cells (0.3x10⁶ cells/mice), and tumors were allowed to grow for 50 days. (**I**) The tumor growth curve of VC +shNT, PITAR OE +shNT, VC +shTRIM28, and PITAR OE +shTRIM28 tumor-bearing mice (n=10) was quantified over time using IVIS. Luminescence was evaluated twice per 10 days and before sacrifice. Bars indicate standard error and the difference between groups was statistically analyzed by ANOVA with Tukey's multiple comparisons post-test (*p<0.05, **p<0.01).

The online version of this article includes the following figure supplement(s) for figure 7:

**Figure supplement 1.** PITAR promotes tumor growth via p53 in a TRIM28-dependent manner.

intervals. In U87/siNT cells, p53 protein peaked around 5 hr of irradiation, while Mdm2 protein peaked at 7 hr of irradiation, as expected (*Figure 6—figure supplement 1E,F and G*; blue line). p21 protein largely followed similar kinetics to that of the Mdm2 protein (*Figure 6—figure supplement 1E and H* ; blue line). In U87/siPITAR#1 cells, while the overall kinetics of p53, Mdm2, and p21 protein were found to be similar in terms of duration, the extent of p53 activation was much stronger (*Figure 6— figure supplement 1E,F,G and H*; red line). The levels of Mdm2 and p21 proteins followed a similarly strong response in *PITAR*-silenced cells (*Figure 6—figure supplement 1E,F,G and H*; red line). From these results and our above findings, we conclude that DNA damage-induced *PITAR* diminishes the p53 response to DNA damage.

## *PITAR* promotes glioma tumor growth in a *TRIM28*-dependent manner and resistance to Temozolomide chemotherapy

To investigate the role of *PITAR* on tumor growth, we checked the ability of *PITAR*-silenced U87 glioma cells to form a tumor in an intracranial orthotopic tumor model using NIH nu/nu mice. We found that *PITAR* silencing (U87/siPITAR) significantly reduced tumor growth (*Figure 7A, B and C*; compare blue line with red line) and enhanced mouse survival (*Figure 7D*). U87/siPITAR tumors showed reduced

TRIM28 staining (*Figure 7E*). Next, we investigated the impact of *PITAR* overexpression on glioma tumor growth and Temozolomide (TMZ) chemotherapy. U87/PITAR OE cells-initiated tumors grew much faster than U87/VC cells (*Figure 7F and G*; compare red line with blue line; *Figure 7—figure supplement 1A*), thus confirming that *PITAR* overexpression promotes tumor growth. While the growth of the tumors formed by U87/VC glioma cells is inhibited substantially by TMZ chemotherapy, U87/PITAR OE tumors showed resistance to TMZ chemotherapy (*Figure 7F and G*; compare pink line with green line; *Figure 7—figure supplement 1A*). U87/PITAR OE tumors showed higher TRIM28 and Ki67 (proliferation marker) staining but reduced p21 staining (*Figure 7—figure supplement 1B*).

To check the importance of TRIM28 in the glioma tumor growth-promoting functions of *PITAR*, we tested the ability of exogenously expressed *PITAR* to promote tumor growth in TRIM28-silenced cells. U87/PITAR OE/shNT cells formed a larger tumor compared to U87/VC/shNT cells (*Figure 7H and I*; compare red line with blue line). However, U87/PITAR OE/shTRIM28 glioma cells formed significantly smaller tumors compared to U87/PITAR OE/shNT cells (*Figure 7H and I*; compare pink line with red line). As expected, small tumors formed by U87/PITAR OE/shTRIM28 glioma cells showed reduced TRIM28 and enhanced p21 staining compared to large tumors formed by U87/PITAR OE/shNT glioma cells (*Figure 7—figure supplement 1C*). These results demonstrate that *PITAR* promotes glioma tumor growth in a TRIM28-dependent manner and confers resistance to TMZ chemotherapy.

To further explore the clinical relevance of our findings, we investigated the survival significance of *PITAR* using the GBM transcriptome datasets. While *PITAR* and *TRIM28* transcripts showed a significant positive correlation in the GBM full cohort and p53 wild-type cohort, there was no significant correlation in the p53 mutant cohort (*Figure 7—figure supplement 1D*). However, survival analysis by univariate Cox regression revealed that *PITAR* transcript level predicted survival only in the p53 wild-type GBM cohort but neither in the full nor p53 mutant cohort (*Figure 7—figure supplement 1E*). The *PITAR* transcript level was dichotomized on further analysis to elucidate the cut-off in predicting the prognosis. We found that high *PITAR* transcript levels predicted poor prognosis significantly compared to low *PITAR* transcript levels in the p53 wild-type cohort (*Figure 7—figure supplement 1G*). However, the *PITAR* transcript levels failed to predict survival in full or p53 mutant GBM cohorts (*Figure 7—figure supplement 1F and H*). These results prove that *PITAR* promotes tumor growth and therapeutic resistance by inactivating p53 by its association with *TRIM28*.

## Discussion

Research into cancer for many decades focussed on protein-coding genes. However, recent evolution in RNA-Seq technologies and bioinformatic methods to analyze transcriptome and genome has changed our perception of noncoding RNA (ncRNA) from 'junk RNAs' to functional regulatory molecules that control various biological processes, such as chromatin remodelling, gene regulation at transcription, post-transcription, and post-translation level, post-translational modifications of proteins, and signal transduction pathways. It appears lncRNAs can influence various macromolecules such as DNA, RNA, and protein to execute specific biological responses and cell fate. Concerning cancer, lncRNAs are critical regulators of cancer and have been shown to influence cancer origin and progression by acting as tumor drivers and/or suppressors in a cancer-specific fashion (*Anastasiadou et al., 2018*; *Slack and Chinnaiyan, 2019*; *Yan and Bu, 2021*). Besides, several lncRNAs are identified as novel biomarkers and potential therapeutic targets for cancer (*Qian et al., 2020*). p53 inactivation is an essential step in cancer development and is associated with therapy resistance. While genetic alteration forms the major mode of p53 inactivation, alternate ways of p53 inactivation, such as amplification of *MDM2* and deletion of *CDKN2A*, have been reported (*Vogelstein et al., 2000*; *Olivier et al., 2002*; *Vousden and Lu, 2002*; *Vousden and Lane, 2007*). Deregulated lncRNAs that modulate p53 activity positively and negatively have been reported (*Jain, 2020*). In this study, we discovered that *PITAR*, an oncogenic lncRNA, inhibits p53 by promoting its ubiquitination through its association with the mRNA of *TRIM28* that encodes p53-specific E3 ligase. *PITAR* is activated by DNA damage in a p53-independent manner, suggesting an incoherent feedforward loop.

Several deregulated lncRNAs acting as oncogenes and tumor suppressors in glioma and glioma stem cells have been identified (*Paul et al., 2018*; *Wang and He, 2019*; *Yadav et al., 2021*). In this study, we identify *PITAR* as a highly expressed lncRNA in GBM and GSCs and display oncogenic properties. *PITAR* is primarily located in the nucleus. Its downregulation by RNA interference inhibited glioma cell growth and colony formation, arrested cells in the G1 phase, induced cell death, and

sensitized cells to chemotherapy. An integrated analysis of *PITAR*-bound targets identified by ChIRP assay and differentially regulated genes in *PITAR* silenced cells identified *PITAR* as an inhibitor p53 through its association with the transcript encoding *TRIM28*.

TRIM28 (KAP1), an E3 ligase, is an Mdm2 interacting protein, and it inhibits p53 through HDAC-mediated deacetylation and direct ubiquitination (*Czerwińska et al., 2017a*; *Wang et al., 2005*). Our results show that the *TRIM28* transcript is upregulated in GBM, as reported earlier (*Czerwińska et al., 2017b*; *Su et al., 2018*), and GSCs in multiple datasets. TRIM28 protein was also found to be high in GBM tissue samples. In addition, we found a significant positive correlation between *PITAR* and *TRIM28*, suggesting that *PITAR* expression is likely to promote *TRIM28* expression. Our results show that *PITAR*-dependent *TRIM28* expression regulation occurs at the post-transcription level via RNA: RNA interactions, where *PITAR* interaction with *TRIM28* mRNA increases its stability. This interaction between *PITAR* and *TRIM28* was confirmed by colocalization using a multiplexed RNAscope. We further demonstrated that the first exon of *PITAR* interacts with 3' UTR of *TRIM28*. Thus, we have comprehensively established that high levels of *PITAR* in GBM promote *TRIM28* expression by binding and stabilizing *TRIM28* mRNA. TRIM28 inhibits p53 through its association with Mdm2 (*Wang et al., 2005*). Since our results show that *PITAR* binding to *TRIM28* transcript promotes *TRIM28* expression, *PITAR* is expected to inhibit p53. Indeed, our results show that *PITAR* inhibits both endogenous and

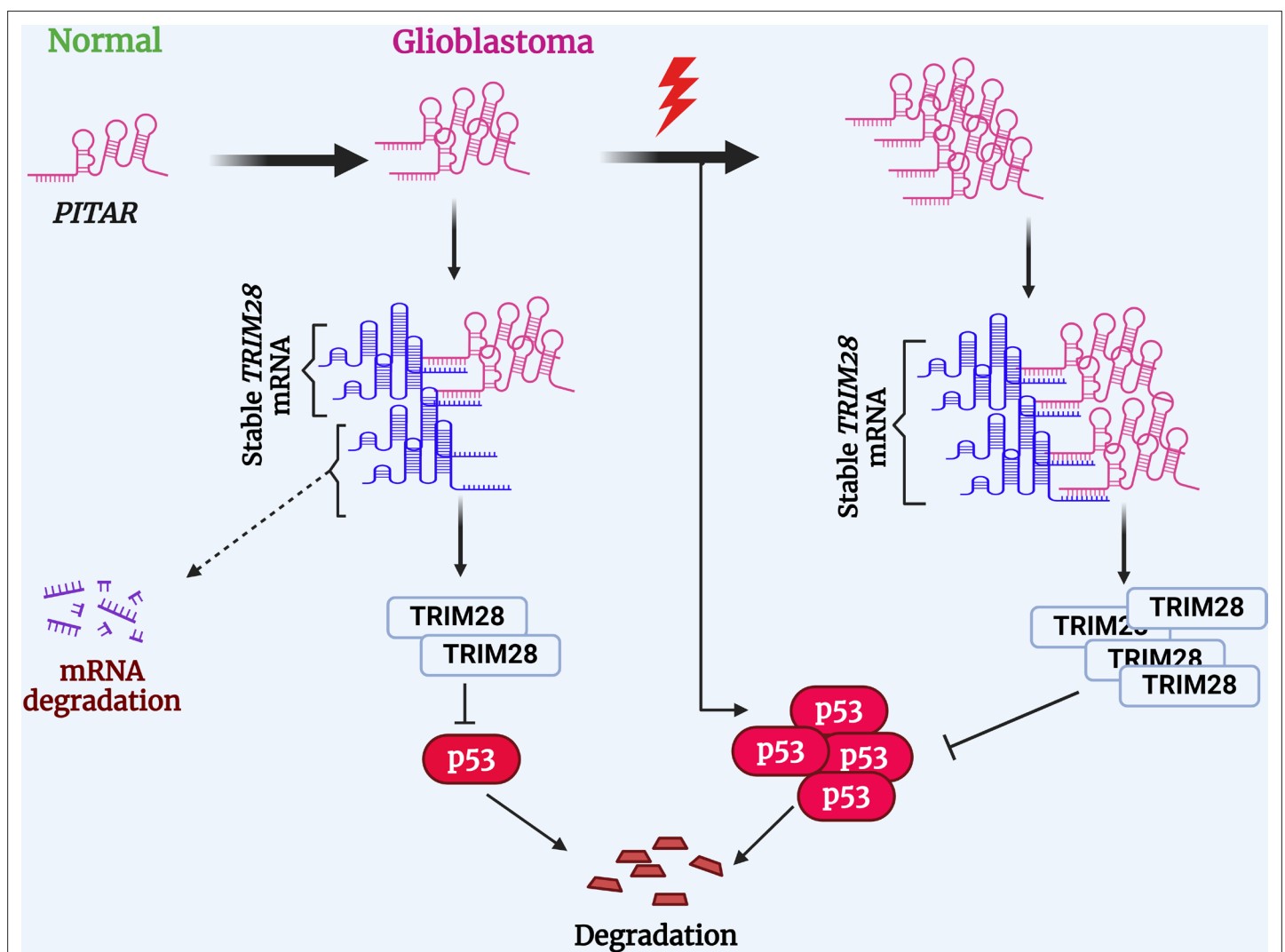

**Figure 8.** Proposed working model of this study. *PITAR* inhibits p53 by binding and stabilizing *TRIM28* mRNA.

DNA damage-induced p53 at the protein level without altering its transcript levels. Increased TRIM28 protein in high *PITAR* cells increased p53 ubiquitination and reduced p53 acetylation. Thus, we establish *PITAR* as a bona fide inhibitor of p53 through its association with *TRIM28* mRNA.

Several cellular stresses, including DNA damage, hypoxia, and nucleotide deprivation, activate p53. Several lncRNAs have been shown to regulate p53 (*Jain, 2020*). LncRNAs such as *LOCC572558*, *NBAT1*, *MT1JP*, *PSTAR*, and *MEG3* are shown to activate p53 using a variety of mechanisms (*Liu et al., 2016b*; *Zhu et al., 2016*; *Uroda et al., 2019*; *Qin et al., 2020*; *Mitra et al., 2021*). In contrast, lncRNAs such as *PURPL*, *MALAT1*, *H19*, and *RMRP* inhibit p53 (*Liu et al., 2016a*; *Chen et al., 2017*; *Chen et al., 2021*; *Li et al., 2017*). While *PITAR* is found to inhibit p53 through its interaction with *TRIM28* mRNA, we also found that *PITAR* level is induced by DNA damage in a p53-independent manner. We show further that DNA damage-induced *PITAR* inhibits the p53 response to DNA damage through its interaction with *TRIM28* mRNA. Thus, based on our data, we propose a model wherein DNA damage not only activates p53, but it also activates *PITAR*, with oncogenic properties, to inhibit p53-dependent functions, thus creating an incoherent feedforward mechanism to attenuate the response of p53 to DNA damage and promote oncogenesis and therapy resistance in GBM (*Figure 8*).

Several oncogenic lncRNAs have been demonstrated as potential therapeutic targets in many cancers (*Anastasiadou et al., 2018*; *Slack and Chinnaiyan, 2019*; *Yan and Bu, 2021*). In GBM, GSCs are the main culprits behind highly aggressive GBM progression (*Visvanathan et al., 2018*). It has been shown that GSCs alone can initiate glioma tumors in mouse models (*Suvà et al., 2014*). Cancer/testis (CT) antigens are a family of tumor-associated antigens expressed only in tumors but not in normal tissues except for the testis and placenta. Their tumor-specific expression and strong in vivo immunogenicity made them suitable for tumor-specific immunotherapy (*Fratta et al., 2011*; *Gjerstorff et al., 2015*). Because of its high expression in GBM and testis normal tissue, *PITAR* is identified as CT lncRNA. *PITAR* silencing efficiently inhibited the growth of glioma cells and GSCs containing WT p53. Intracranial orthotopic mouse model experiments show that *PITAR* promotes glioma tumor growth and confers resistance to temozolomide chemotherapy. Our study establishes *PITAR* as an oncogenic CT lncRNA, which promotes oncogenesis by inactivating p53 by interacting with *TRIM28* mRNA. Thus, it can serve as a potential therapeutic target for GBM.

## Materials and methods
### Tumor samples and clinical details
Tumor samples were collected from patients operating at the hospital (NIMHANS, Bangalore). The tissue samples were obtained with written consent from all patients before being used in the current study. The study has been approved by the ethics committee of the NIMHANS and the Indian Institute of Science (IISc). A portion of the anterior temporal cortex resected during surgery for drug-resistant epilepsy patients served as a control brain sample (NIMHANS, Bangalore). We used GBM (n=79) and Normal (n=3) tissue to quantify *PITAR* and *TRIM28* expression and correlation. The neuropathologist confirmed the histology as GBM, IDH wild type.

### Cell lines
Primary human tumor-derived GSCs Neurosphere (MGG4, MGG6, MGG8) are obtained from Wakimoto's Lab, RG5 and 120–53 GSCs Neurosphere obtained from Sánchez-Gómez's Lab, and other GSCs Neurosphere (GB1, GB2, GB3) are developed in our Lab. The Neurospheres were grown in Neurobasal medium (#21103049, Gibco) supplemented with l-glutamine, heparin, B27 supplement, N2 supplement, rhEGF, rhFGF-basic, penicillin, and streptomycin. To make single-cell suspensions for re-plating, the spheres were chemically dissociated after seven days of plating using the NeuroCult Chemical Dissociation Kit (mouse, #05707) from Stem Cell Technologies, according to the manufacturer's instructions. The monolayer culture GBM cell lines U87, U343, U252, LN229, A172, and U373 were obtained from The European Collection of Authenticated Cell Cultures (ECACC) and immortalized human astrocyte cell line IHA (NHA-hTERT-E6/E7) was obtained from Dr. Russell Pieper's laboratory, University of California, San Francisco (San Francisco, CA), and cells are maintained in specific culture media with 5% CO2 and humidified incubator at 37°C. The mycoplasma contamination was tested using RT–PCR. All cell lines are verified to be mycoplasma-free by EZdetect PCR Kit for Mycoplasma Detection (HiMedia).

## Plasmids

The shRNA plasmid (pLKO.1) for *TP53* (TRCN0000003754) and *TRIM28* (TRCN0000017998) were obtained from the TRC library (Sigma, IISc). The TRIM28 3' UTR Luc vector plasmid was purchased from Origen, USA (# SC202088). PG13-Luc vector plasmid was obtained from Addgene, USA (#16442). The Flag-TRIM28 overexpression plasmid was bought from Addgene, USA (# 124960). The pRK5-HA-Ubiqitin-K48 plasmid was procured from Dr. Sashank Tripathi's laboratory (Addgene_17605).

## Antibodies and reagents

Primary antibodies were purchased from the following commercial vendors: p53 (DO-1) from Santacruz; acetylated p53 (K382), p21, TRIM28, and Anti-Ub from Cell Signaling Technology; GAPDH and β-actin from Sigma Aldrich. Goat anti-mouse HRP conjugate (Bio-Rad #170–5047, WB 1:5000), goat anti-rabbit (H+L) secondary HRP conjugate (Invitrogen, #31460, WB 1:5000), goat anti-mouse IgG (H+L) highly cross-adsorbed secondary antibody, Alexa Fluor 488 (Invitrogen, #A-11029), goat anti-rabbit IgG (H+L) highly cross-adsorbed secondary antibody, Alexa Fluor 488 (Invitrogen, Cat# A-11034), goat anti-mouse IgG (H+L) highly cross-adsorbed secondary antibody, Alexa Fluor 594 (Invitrogen, Cat# A-11032), goat anti-rabbit IgG (H+L) highly cross-adsorbed secondary antibody, Alexa Fluor 594 (Invitrogen, Cat#A-11037). All Alexa Fluor conjugated antibodies were used at a dilution of 1:500 for IHC and ICC. The PCR primers were purchased from Sigma, and the Biotin-TEG DNA antisense oligos were designed in *Stellaris probe* designer and purchased from IDT. *PITAR* siRNAs and negative control oligos were purchased from Eurofins. Doxorubicin, Temozolomide, CGK733, MG132, Actinomycin D, and Cycloheximide were purchased from Sigma.

## Plasmid construction and RNA interference

The PITAR OE partial clone (~1846 bp) was made in the pCDNA3.1 vector backbone, and the deletion clone (ΔPITAR OE) was made using a Q5 site-directed mutagenesis assay kit (# E0554S, NEB). The deletion clone of *TRIM28* 3'UTR (ΔTRIM28 3'UTR) was made using a Q5 site-directed mutagenesis assay kit. The *PITAR* Knockdown was performed using *PITAR* siRNA and transfected using Dharmafect1 transfection reagent. Cells were harvested at the 24th, 48th, 72nd, and 96th hr post-transfection to check for knockdown of desired genes at mRNA level by qRT-PCR.

To prepare the Lentivirus for shRNA (pLKO.1 vector), HEK-293T cells were transfected with shRNA plasmid and helper plasmids psPAX2 and pMD2.G using Lipofectamine 2000 (Invitrogen #11668027) in Opti-MEM (Invitrogen #22600–050) medium. After 6 hr of transfection, the Opti-MEM medium was replaced by a fresh DMEM supplemented with 10% FBS and the virus was collected after 60 hr of transfection. The knockdown of p53 and TRIM28 was performed using shTP53 and shTRIM28 lentivirus infection, followed by puromycin selection. The exogenous expression of p53 was executed by using a recombinant adenovirus for p53 and compared to the control virus (Ad-GFP) which is amplified in HEK-293 cells.

## List of siRNA and shRNA used in this study

| No. | Name | 5' Sequence 3' |
|-----|------|----------------|
| 1 | siPITAR#1 | GAAGCAUCCUUCCUGAUCAdTdT |
| 2 | siPITAR#2 | CAGACUUCCUGUACUACCUdTdT |
| 3 | siPITAR#3 | GUAGCAAGAAGAGGUCUCAdTdT |
| 4 | shp53 #1 | CCGGGTCCAGATGAAGCTCCCAGAACTC GAGTTCTGGGAGCTTCATCTGGACTTTTT |
| 5 | shp53 #2 | CCGGCACCATCCACTACAACTACATCTCG AGATGTAGTTGTAGTGGATGGTGTTTTT |
| 6 | shTRIM28 #1 | CCGGCCTGGCTCTGTTCTCTGTCCTCTCGAGA GGACAGAGAACAGAGCCAGGTTTTT |
| 7 | shTRIM28 #2 | CCGGCCTGGCTCTGTTCTCTGTCCTCTCGAGA GGACAGAGAACAGAGCCAGGTTTTT |

## The identification of differentially expressed lncRNA and mRNA in GBM vs. control brain tissue and GSC vs. DGC

The Raw RNA sequencing data was obtained for GBM samples from TCGA (https://tcga-data.nci.nih.gov/tcga/). The whole RNA sequencing data were aligned using the PRADA tool (*Torres-García et al., 2014*). Duplicate removal was carried out using Picard 1.73 (http://broadinstitute.github.io/picard/; *Broad Institute, 2019*), and the lncRNAs were annotated as per the Gencode Version 19 annotation file (https://www.gencodegenes.org/human/release_19.html). The RNA-seq data for GSC vs. DGC was obtained from GSE54791 (*Suvà et al., 2014*). The gene expression matrix obtained was log2 transformed. Further, the count expression matrix for GBM and Normal was obtained from TCGA, CPM was normalized using library edgeR, and limma using R. Common protein-coding genes and lncRNA were chosen, and a scatterplot was constructed among them. The pan-cancer data for the gene expression profile was derived from the TCGA browser (https://tools.altiusinstitute.org/tcga/?gene=FAM95B1). The tissue-specific gene expression and regulation were analyzed using the GTEx portal (Samples were collected from 54 non-diseased tissue sites across nearly 1000 individuals). The quantitative proteomics data of tumors was derived from the CPTAC data portal.

## RNA-sequencing

Total RNA was extracted from siNT and siPITAR cells using the Trizol method (QIAGEN). RNA quality was assessed using an Agilent TapeStation system, and a cDNA library was made. According to published protocols, each sample was sequenced using the Illumina HiSeq 2000 (with a 100-nt read length). To quantitate the abundance under this condition. The raw reads were quality-analyzed, and upon satisfactory assessment, Kallisto was used with default parameters to quantitate and obtain TPM-normalized gene abundance. The fold change was brought using Deseq2. IGV was used to visualize the raw reads. David and GSEA were performed both at log2fold 0.58 and p<0.05. The gene expression matrix between siPITAR and Control was used to construct a volcano plot to visualize differentially expressed genes.

## Gene set enrichment analysis (GSEA)

The differentially expressed genes between the siNT and siPITAR (as identified from RNA-seq) were pre-ranked based on fold change and used as an input to perform GSEA. All the gene sets available in the Molecular Signature Database (MSigDB, roughly 18,000 gene sets) were used to run the GSEA. We filtered out the cell cycle, apoptosis, and p53 pathway-related gene sets to identify that most of them were significantly enriched in the siPITAR over the siNT. We acknowledge using the GSEA software and MSigDB (http://www.broad.mit.edu/gsea/; *Subramanian et al., 2005*).

## RNA isolation and real-time quantitative RT-PCR analysis

Total RNA was isolated using TRI reagent (Sigma, U.S.A.), and 2 µg of RNA was reverse transcribed with the High-capacity cDNA reverse transcription kit (Life Technologies, USA) according to the manufacturer's protocol. qRT-PCR was performed using DyNAmo ColorFlash SYBR Green qPCR Kit in the ABI Quant Studio 5 Sequence Detection System (Life Technologies, USA). The expression of the genes of interest was analyzed by the ΔΔCt method using ATP5G and GAPDH as internal control genes. Real-time primer information is provided below.

## List of primers used in this study

| No. | Primers | 5' Sequence 3' |
|---|---|---|
| qRT-PCR Primers | | |
| 1 | PITAR_Exn1 Forward | CCAGGGTCTGCTTAGAGAGG |
| 2 | PITAR_Exn1 Reverse | AGGCTACCACTAAGCCACAG |
| 3 | PITAR Forward | CGACCTGGTGCACAACTTTA |
| 4 | PITAR Reverse | CTCAGCACAAACGCATCACT |

*Continued on next page*

*Continued*

| No. | Primers | 5' Sequence 3' |
| --- | --- | --- |
| 5 | TP53 Forward | CCTCAGCATCTTATCCGAGTGG |
| 6 | TP53 Reverse | TGGATGGTGGTACAGTCAGAGC |
| 7 | MDM2 Forward | TGTTTGGCGTGCCAAGCTTCTC |
| 8 | MDM2 Reverse | CACAGATGTACCTGAGTCCGATG |
| 9 | CDKN1A Forward | AGGTGGACCTGGAGACTCTCAG |
| 10 | CDKN1A Reverse | TCCTCTTGGAGAAGATCAGCCG |
| 11 | GAPDH Forward | GTCTCCTCTGACTTCAACAGCG |
| 12 | GAPDH Reverse | ACCACCCTGTTGCTGTAGCCAA |
| 13 | TRIM28 Forward | CAAGATTGTGGCAGAGCGTCCT |
| 14 | TRIM28 Reverse | CATAGCCTTCCTGCACCTCCAT |
| 15 | RPL35 Forward | TGCCCGTGTTCTCACAGTTA |
| 16 | RPL35 Reverse | CAGGGGCTTGTACTTCTTGC |
| 17 | ATP5G Forward | ACAGCAACTTCCCACTCCAG |
| 18 | ATP5G Reverse | ACTTGGCTGCTGTGTCAATG |
| 19 | TRIM28 3'UTR_Forward | CAGGAGCTGTCTGGTGGC |
| 20 | TRIM28 3'UTR_Reverse | GAGTGGGGATGGGGTGAC |
| Site-directed Mutagenesis primer | | |
| 21 | Q5SDM PITAR_Exn1_Forward | AGGTCGGACCCTGTGAGG |
| 22 | Q5SDM PITAR_Exn1_Reverse | ATAGACTGTTACCATCTCTCTAGCC |
| 23 | Q5SDM TRIM28_Forward | CATCCCCCAGTTCCTCACGATATG |
| 24 | Q5SDM TRIM28_Reverse | CTGGCCATGGGGGCTCCA |
| Invitro Transcription primer | | |
| 25 | T7_PITAR_ Forward | TAATACGACTCACTATAGGGCTGCC ATAGTGGAAGTTTCTC |
| 26 | T7_PITAR_ Reverse | TAATACGACTCACTATAGGGCTAAG GCAACCAAGGCAGAG |
| 27 | TRIM28 3'UTR _Forward | TAATACGACTCACTATAGGGCTGAG TTCCCAGGAGCTGTC |
| 28 | TRIM28 3'UTR _Reverse | ATACAGTCAATAAACCAGGC |
| Cloning primer | | |
| 29 | PITAR_ Exon1 Forward | CCCAAGCTTCTGCCATAGTGGAAGTTTCTC |
| 30 | PITAR_ Exon1 Reverse | CCGCTCGAGCTAAGGCAACCAAGGCAGAG |

## Cytosolic/nuclear fractionation

Cells were incubated with hypotonic buffer (25 mM Tris–HCl (pH 7.4), 1 mM MgCl$_2$, 5 mM KCl, and RNase inhibitor) on ice for 5 min. An equal volume of hypotonic buffer containing 1% NP-40 was added, and the sample was left on ice for another 5 min. After centrifugation at 5000 × *g* for 5 min, the supernatant was collected as the cytosolic fraction. The pellets were resuspended in nuclear resuspension buffer (20 mM HEPES (pH 7.9), 400 mM NaCl, 1 mM EDTA, 1 mM EGTA, 1 mM dithiothreitol, 1 mM phenylmethyl sulfonyl fluoride and RNase inhibitor) and incubated at 4 °C for 30 min. The nuclear fraction was collected after removing insoluble membrane debris by centrifugation at 12,000 × *g* for 10 min. RNA isolation was performed using the Trizol method.

## RNA stability assays

U87 cells were treated with 5 µg/ml actinomycin D at various times as indicated (0, 60, 120 min). RNA was extracted, cDNA was made, and qRT–PCR was carried out as described above. RNA half-life (t1/2) was calculated by linear regression analysis.

## ChIRP assay

The targets of LncRNA were identified using ChIRP-RNA sequencing, which was described earlier (*Chu et al., 2012*). Antisense DNA probes for *PITAR* were designed using the Stellaris Probe Designer tool. Probes were labeled with Biotin-TEG at the 3′ ends. U87 cells were crosslinked with 1% glutar-aldehyde for 10 min at 37 °C and then quenched with 0.125 M glycine buffer for 5 min. U87 cells were lysed in lysis buffer (50 mM Tris, pH 7.0, 10 mM EDTA, 1% SDS, DTT, PMSF, protease inhibitor, and RNase inhibitor) on ice for 30 min, and genomes were sonicated three times into fragments 300–500 bp in length. Chromatins were diluted twice the volume of hybridization buffer (750 mM NaCl, 1% SDS, 50 mM Tris, pH 7.0, 1 mM EDTA, 15% formamide, DTT, PMSF, protease inhibitor and RNase inhibitor). Biotin-TEG labeled probes (odd, even, and LacZ) were added, and mixtures were rotated at 37 °C for 4 hr. Streptavidin-magnetic C1 beads were blocked with 500 ng/µl yeast total RNA and 1 mg/ml BSA for 1 hr at 25 °C and washed three times before use. We incubated biotin probes with U87 cell lysates and then used Streptavidin C1 magnetic beads for capture. Finally, beads were resolved for RNA by the RNA elution buffer. The eluted RNA was subjected to RNA sequencing. The raw reads were quality-analyzed to quantify the abundance of mRNA in the ChIRP assay. Upon satisfactory assessment, Kallisto was used with default parameters to quantitate and obtain TPM-normalized gene abundance. IGV was used to visualize the raw reads. The Biotin-TEG labeled Anti-sense probe sequences are provided below.

## List of probes used in this study

| No. | ChIRP Probe (Biotin-TEG) | 5′ Sequence 3′ |
|---|---|---|
| 1 | PITAR Antisense_1 | CACCAAGACCTGCACTACTC |
| 2 | PITAR Antisense_2 | TGACAAGGCTACCACTAAGC |
| 3 | PITAR Antisense_3 | AGACTGTTACCATCTCTCTA |
| 4 | PITAR Antisense_4 | GACACTTGAAAAGCGGGACC |
| 5 | PITAR Antisense_5 | TTTTCTGAGTCCTGAGACAG |
| 6 | PITAR Antisense_6 | CTTGCCACAAAATGTGCACA |
| 7 | PITAR Antisense_7 | CACAATAGCAGTTCTGGGTT |
| 8 | PITAR Antisense_8 | GATTCCTGGAGGGAACCTTG |
| 9 | PITAR Antisense_9 | CCAGATTTCTTCTGGTCATT |
| 10 | PITAR Antisense_10 | CAGGTAAGGACAGTGTGCTA |
| 11 | PITAR Antisense_11 | ACCAAGAGACAACCCCTAAC |
| 12 | PITAR Antisense_12 | GTTTAGGTTTACCTAGGACT |
| 13 | PITAR Antisense_13 | AAATGGGACTCCCTTGTAGA |
| 14 | PITAR Antisense_14 | CTGTGCTGTCATATCCTAAG |
| 15 | LacZ_1 | TCACGACGTTGTAAAACGAC |
| 16 | LacZ_2 | ATTAAGTTGGGTAACGCCAG |
| 17 | LacZ_3 | AGGTTACGTTGGTGTAGATG |
| 18 | LacZ_4 | AATGTGAGCGAGTAACAACC |
| 19 | LacZ_5 | GTAGCCAGCTTTCATCAACA |
| 20 | LacZ_6 | AATAATTCGCGTCTGGCCTT |

*Continued on next page*

*Continued*

| No. | ChIRP Probe (Biotin-TEG) | 5' Sequence 3' |
|-----|--------------------------|----------------|
| 21 | LacZ_7 | AGATGAAACGCCGAGTTAAC |
| 22 | LacZ_8 | AATTCAGACGGCAAACGACT |
| 23 | LacZ_9 | TTTCTCCGGCGCGTAAAAAT |
| 24 | LacZ_10 | ATCTTCCAGATAACTGCCGT |
| 25 | LacZ_11 | AACGAGACGTCACGGAAAAT |
| 26 | LacZ_12 | GCTGATTTGTGTAGTCGGTT |

## Immunoblotting and co-immunoprecipitation

The western blot analysis was described earlier (*Fong et al., 2018*). In brief, RIPA buffer with protease inhibitor cocktail was used to isolate protein from the GBM cell lines, and GSCs were quantified by Bradford's reagent. These studies used the following antibodies: p53, p21, GAPDH, anti-Ub, and TRIM28. To measure the stability of the p53 protein, the cells were treated with Cycloheximide at a final concentration of 50 µg/ml, and immunoblotting was performed from whole cell lysates.

The endogenous ubiquitination assay was described earlier (*Fong et al., 2018*). Briefly, we transfect the HA-Ubiquitin in VC/PITAR OE stable U87 cells and performed co-immunoprecipitation with anti p53 antibody to detect the ubiquitination of the p53 protein; MG132 treated and untreated cells were lysed in IP lysis buffer (0.5% NP-40, 150 mM NaCl, 20 mM HEPES, pH 7.4, 2 mM EDTA, and 1.5 mM MgCl2 and 20 mM Iodoacetamide (IAA)) supplemented with protease inhibitor cocktail for half an hour on ice. Cell lysates were incubated with protein G-Magnetic beads (Dynabeads Protein G beads) coated with the p53 antibodies overnight at 4 °C, and then the IP products were washed three times with IP lysis buffer; after that, the proteins were eluted with SDS sample buffer, and the eluted protein were run on SDS–PAGE followed by immunoblotting and probed with anti-ubiquitin antibody.

## Luciferase reporter assay

Luciferase assays were performed using reporter lysis buffer (Catalog #E3971, Promega, USA) and luciferase assay reagent according to the manufacturer's instructions. Briefly, plasmids were transfected in the cells plated in 12 well plates. For determining the comparative luciferase activity of PITAR OE/ ΔPITAR OE plasmid was co-transfected with TRIM28 3'UTR luc plasmid/ ΔTRIM28 3'UTR luc plasmid. Cells were harvested after 48 hr of transfection, and lysates were made. The luciferase assays were performed using a luciferase assay substrate (Catalog #E151A, Promega, USA), and luciferase readings were recorded using a luminometer (Berthold, Germany) using an equal quantity of protein measured by Bradford assay. ß-Gal assays were performed to normalize the transfection differences of the PG13 Luc assay, and RFP fluorescence intensity was measured to normalize TRIM28 3'UTR Luc activity.

## In vitro transcription

The plasmid DNA template is used for in vitro synthesis of biotinylated PITAR/ ΔPITAR sense, antisense, generated by PCR amplification. The forward primer contained the T7 RNA polymerase promoter sequence, allowing for subsequent in vitro transcription. PCR products were purified using the DNA Gel Extraction Kit (Thermo Fisher) and used as templates for synthesizing biotin-labeled probes. The reaction was carried out using RiboMAX Large Scale RNA Production System (T7) (Catalog #P1300, Promega, USA) and biotin-11-UTP (Catalog #AM8451, Invitrogen, USA) as per manufacturer's instructions. Briefly, 2.5 µg of linear template DNA was added, and the reaction was incubated at 37 °C for 1.5 hr. After alcohol precipitation, the RNA was resuspended in 20 µl of nuclease-free water. Primers used for amplifying the region from the PITAR OE construct with T7 promoter overhang are provided above.

## Biotinylated RNA pulldown assay

The in vitro transcribed RNA was treated with DNase (Ambion) and purified with an RNeasy kit (QIAGEN). Twenty pmol biotinylated RNA (PITAR-Antisense, PITAR-Sense, and ΔPITAR-Sense) was incubated with 1 mg whole lysate prepared from U87 cells in binding buffer [10 mM HEPES, pH 7.4, 100 mM KCl, 3 mM MgCl$_2$, 5% glycerol, 1 mM DTT, Yeast transfer RNA (50 ng/ml)] and heparin (5 mg/ml) for 4 hr at 4 °C. The biotinylated RNA-RNA complexes were pulled down by incubation with Dynabeads M-280 Streptavidin (Thermo Fisher Scientific) for 4 hr at 4 °C. After brief centrifugation, bound RNA in the pulldown material was separated by elution followed by the TRI method. Afterward, RNA was converted to cDNA and quantified by using qRT-PCR. In these assays, a nonspecific transcript derived from the T7 promoter-luciferase construct and bead alone was added with lysate as a control for nonspecific binding.

## Antisense oligo blocking followed by ChIRP assay

To investigate the *TRIM28* interaction site on the *PITAR* transcript, we performed a competition ChIRP assay. In brief, the fragmented glutaraldehyde fixed RNA was preincubated with biotin-less antisense oligo number three, located at the energetically favorable binding site of the *TRIM28* transcript. Next, we incubated with a biotin-labeled odd antisense probe set and performed a ChIRP pulldown assay. The interacting RNA was eluted, and we performed cDNA preparation followed by qRT-PCR to detect the interacting *TRIM28* transcript.

## Transfection of GSCs, neurosphere assay, and sphere diameter measurement

The transfection in GSCs was carried out in a single-cell suspension state for siRNA and plasmid DNA. 100 nM concentration of either Control non-targeting siRNA or gene-specific siRNA (Dharmacon, UK), as indicated, were transfected using Dharmafect I (Dharmacon, U.K) according to the manufacturer's instructions. Control vector or PITAR OE plasmids were transfected using Lipofectamine 2000 (Life Technologies, USA) according to the manufacturer's instructions. After 72 hr of transfection, cells were harvested and confirmed for gene manipulation by qRT-PCR. After 48 hr of transfection, the aggregates formed were dissociated into single cells, counted, and equal numbers of cells were plated at a density of 2 cells/µl in 24-well or six-well plates. The number of spheres was counted after 7 days of plating. Fresh medium was replenished every 2–3 days. Sphere diameter measurements were analyzed using ImageJ software. The number of spheres above 50 µM diameter was counted and plotted for the total number of spheres.

## Limiting dilution assay

For each condition, GSCs (single cells) were plated 1, 10, 50, 100, and 200 cells in 10 wells each, respectively, of a 96-well plate, and sphere formation was assessed over the next 5–7 days. The number of wells not forming spheres was counted and plotted against the number of cells per well. Extreme limiting dilution assay was done using the online ELDA software (https://bioinf.wehi.edu.au/software/elda/; *Hu and Smyth, 2009*).

## Xenograft orthotopic mouse model

The experiments were conducted using Athymic Nude female mice (6–8 weeks old), with approval from the Institute Ethical Committee for Animal Experimentation (IAEC approval No: CAF/Ethics/692/2019). The mice were housed under a 12 hr light/dark cycle, provided with a regular diet ad libitum, and the experiments were carried out during the light phase of the cycle. U87MG-Luc cells bearing Control siNT/VC/shNT or siPITAR#1/PITAR OE/shTRIM28 (0.3×10$^6$) were intracranially injected into the right corpus striatum by stereotactic injection 3 mm deep of 6- to 8-week-old immunocompromised CD1 nu/nu female mice (IISc, CAF; n=10 mice per group). The mice were kept in a 12 hr light and dark cycle, fed ad libitum with a regular diet, and the experiments were done in the light phase of the cycle. Intracranial tumors were monitored by bioluminescence imaging with the PerkinElmer IVIS Spectrum using mild gas anesthesia (using isoflurane) for the animals, and total photon flux (photons/s) was measured every 5–6 days intervals and plotted. Temozolomide (100 mg/kg BW) was administered intraperitoneally every day for 1 week. After 4 weeks of tumor injection, mice were sacrificed. The

survival of mice in both groups (siNT and si PITAR#1, n=6) was followed up, and the survival curve was plotted using GraphPad Prism.

## RNAScope

RNAScope in situ hybridization (ISH) was performed using RNAscope Multiplex Fluorescent Reagent Kit v2 (Cat. No. 323100) from Advanced Cell Diagnostics (ACD), which was described earlier (*Mahale et al., 2022*). RNA probes specific to *PITAR* and *TRIM28* were designed by ACD's made-to-order probes service. Manual RNAScope ISH protocol from Advanced Cell Diagnostics (ACD, 323100-USM) was followed to perform single or double RNA staining. RNAScope manual procedure involved sample fixation, sample pretreatment, subsequent probe hybridization, signal amplification, and ISH signal detection. RNA in situ (RNAScope) and immunofluorescence imaging were performed on a Zeiss LSM 880 airy scan confocal microscope at the Centre for Cellular Imaging Facility. Most images were captured on 40 X or 60 X oil immersion objectives with the laser emitting 405, 488, 561, and 670 nM wavelengths, depending on the fluorophores used in the experiments. The 3D reconstruction was performed using Imaris microscopy image analysis software 9.8.2 version.

## Cell viability

The Vi-cell reagent Kit (Beckman Coulter) was used according to the manufacturer's instructions. Briefly, cells were seeded at $1 \times 10^5$ per well in 12-well plates overnight before treatment as desired. Cells are harvested and resuspended in 1 ml 1 X PBS. The viable cell count was done in the Vi-Cell cell viability Analyzer (Beckman Coulter).

## Apoptosis

Apoptotic cells were quantitated using the Annexin V Apoptosis Detection Kit (BD Biosciences). In brief, cells were washed twice with cold PBS and then resuspended in binding buffer at a concentration of $1 \times 10^6$ cells per ml. 100 µl of the solution ($1 \times 10^5$ cells) was transferred to a 5 ml culture tube, and 5 µl of annexin V was added. After incubation at room temperature for 15 min in the dark, an additional 400 µl of binding buffer was added to each tube, and cells were analyzed using a flow cytometer within 1 hr (FACSVerse, BD Biosciences).

## Cell cycle analysis

Cells were fixed by 70% ethanol at –20 °C overnight and spun down at 4000 r.p.m. Cell pellets were resuspended in PBS containing 0.25% Triton X-100 and incubated on ice for 15 min. After discarding the supernatant, the cell pellet was resuspended in 0.5 ml PBS containing 10 µg/ml RNase A and 20 µg/ml propidium iodide stock solution and incubated at room temperature in the dark for 30 min. Cells were then subjected to analysis using a flow cytometer (FACSVerse).

## Colony formation assay

U87 and U343 cells were transfected with the *PITAR* siRNA and control siRNA. Twenty-four hours later, $1 \times 10^3$ cells were cultured in a 6-well plate. Two weeks later, cells were fixed, stained with crystal violet, and photographed. The percentage and intensity of the area covered by crystal violet-stained cell colonies were quantified using the ImageJ software.

## Fluorescent immunohistochemistry

The FFPE tissue sections (5 µm thick) were de-waxed and rehydrated. Antigen retrieval was performed in a pressure cooker for 20 min in 10 mM Tris with 1 mM EDTA (pH 9). Nonspecific binding was blocked using blocking buffer (PBS (pH 7.4), 3% serum, 1% BSA, and 0.1% Tween) for 60 min at room temperature. Sections were then incubated with primary antibodies (Ki67, TRIM28, and p21) diluted in blocking buffer overnight at 4 °C. After washing twice with 0.1% PBS–Tween, slides were incubated with a secondary antibody conjugated with Alexa 488 (Thermo Fisher). After washing, sections were incubated with DAPI (Sigma-Aldrich) as a counterstaining. The sections were mounted using prolonged antifade glass mounting media (Thermo Fisher). Two investigators examined the slides. The percentage of positive cells was estimated from 0% to 100%. The intensity of staining (intensity score) was judged on an arbitrary scale of 0–4: no staining (0), weakly positive staining (1), moderately

positive staining (2), strongly positive staining (3), and very strong positive staining (4). An immunoreactive score was derived by multiplying the percentage of positive cells with staining intensity divided by 10.

## H&E staining

Brain tissues were fixed with 4% paraformaldehyde, dehydration (gradient ethanol), and embedding in paraffin. Then, the Brain tissues were cut into 5 µm slices using a microtome instrument. Afterward, the slices were dewaxed with xylene I and xylene II, dehydrated with 95%, 90%, 80%, and 70% ethanol, and addressed with distilled water. Finally, the slices were processed with Harris hematoxylin, 1% hydrochloric acid alcohol, 0.6% ammonia, and eosin. After dehydration (gradient ethanol) and immersion (xylene), the Brain tumor's pathologic structure was observed with a microscope.

## Statistical analysis

Statistical analyses were performed using GraphPad Prism 6 (GraphPad Software, La Jolla, CA) or R software. Unpaired t-tests or one-way ANOVA, followed by t-tests for individual group comparisons (Tukey's test), were used as described for each experiment. Data are presented as either means ± s.e.m. or means ± s.d. All the experiments were performed in biological triplicate unless otherwise specified. $p < 0.05$ was considered to be statistical significance.

## Acknowledgements

The results published here are, in whole or part, based upon data generated by The Cancer Genome Atlas pilot project established by the NCI and NHGRI. Information about TCGA and the investigators and institutions that constitute the TCGA research network can be found at http://cancergenome.nih.gov/. We acknowledge the shRNA consortium (Dr. Subba Rao), IISc, India, for shRNA constructs. SJ acknowledges NPDF, DBT RA, and DST for fellowship. BG acknowledges DST Inspire, KRP acknowledges DBT for fellowship. CK was supported by grants from the Swedish Cancer Research Foundation [Cancerfonden: Kontrakt no. 211416Pj01H]; Swedish Research Council [2022–01262]; Barncancerfonden [PR2021-0026]; Ingabritt och Arne Lundbergs forskningsstiftelse (LU2020-0017), LUA/ALF and the VAJRA Faculty Scheme grant (VJR/2021/000008) from the Department of Science and Technology, Government of India. KS acknowledges DBT, and MHRD-STARS (Govt. of India) for research grants. Infrastructure supported by DST FIST, DBT-IISc partnership program, UGC, and IOE to IISc is acknowledged. KS is awarded JC Bose Fellowship from DST.

## Additional information

### Competing interests

Kumaravel Somasundaram: Reviewing editor, *eLife*. The other authors declare that no competing interests exist.

### Funding

| Funder | Grant reference number | Author |
| --- | --- | --- |
| Department of Biotechnology, Ministry of Science and Technology, India | | Kumaravel Somasundaram |
| Ministry of Human Resource Development | STARS | Kumaravel Somasundaram |
| Department of Science and Technology, Ministry of Science and Technology, India | NPDF | Samarjit Jana |
| Swedish Cancer Research Foundation | 211416Pj01H | Chandrasekhar Kanduri |

| Funder | Grant reference number | Author |
|---|---|---|
| Swedish Research Council | 2022–01262 | Chandrasekhar Kanduri |
| Barncancerfonden | PR2021-0026 | Chandrasekhar Kanduri |
| IngaBritt och Arne Lundbergs Forskningsstiftelse | LU2020-0017 | Chandrasekhar Kanduri |
| LUA/ALF and the VAJRA Faculty Scheme | (VJR/2021/000008 | Chandrasekhar Kanduri |

The funders had no role in study design, data collection and interpretation, or the decision to submit the work for publication.

### Author contributions

Samarjit Jana, Conceptualization, Data curation, Formal analysis, Funding acquisition, Validation, Investigation, Visualization, Methodology, Project administration, Writing - review and editing; Mainak Mondal, Sagar Mahale, Kaval Reddy Prasasvi, Data curation; Bhavana Gupta, Lekha Kandasami, Neha Jha, Abhishek Chowdhury, Formal analysis; Vani Santosh, Resources, Writing - review and editing; Chandrasekhar Kanduri, Resources, Investigation, Methodology, Writing - review and editing; Kumaravel Somasundaram, Conceptualization, Resources, Supervision, Funding acquisition, Investigation, Visualization, Writing - original draft, Project administration, Writing - review and editing

### Author ORCIDs

Samarjit Jana http://orcid.org/0000-0002-8743-4711
Abhishek Chowdhury https://orcid.org/0009-0007-2985-9053
Kumaravel Somasundaram https://orcid.org/0000-0001-6228-9741

### Ethics

The tissue samples were obtained with written consent from all patients before being used in the current study. The study has been approved by the ethics committee of the NIMHANS and the Indian Institute of Science (IISc).

Experiments were performed in Athymic Nude female mice (6-8 weeks old) following the approval by the Institute Ethical Committee for Animal Experimentation (Institute Animal Ethics Committee [IAEC] Project Number: CAF/Ethics/752/2020).

Reviewer #1 (Public review): https://doi.org/10.7554/eLife.88256.3.sa1
Reviewer #2 (Public Review): https://doi.org/10.7554/eLife.88256.3.sa2
Author response https://doi.org/10.7554/eLife.88256.3.sa3

## Additional files

### Supplementary files

• Supplementary file 1. Differentially expressed Genes in GSC vs. DGC derived from GSE54791 and GBM vs. Normal derived from TCGA.

• Supplementary file 2. ChIRP enriched genes in Even, Odd and LacZ pulldown.

• Supplementary file 3. Differentially regulated genes (GBM VS Control brain) that correlate with PITAR expression.

• Supplementary file 4. Upregulated Genes in GBM – Normal from TCGA dataset.

• Supplementary file 5. Upregulated Genes in GSC – DGC from GSE54791 dataset.

• Supplementary file 6. Differentially expressed Genes in siPITAR/U87 - siNT/U87.

• Supplementary file 7. GSEA analysis of downregulated genes in siPITAR/U87 - siNT/U87.

• Supplementary file 8. In-silico prediction of RNA-RNA interaction site between TRIM28 and PITAR using IntaRNA tools.

• MDAR checklist

## Data availability

The newly genareted Sequencing data (ChIRP-RNA-seq and siPITAR RNA-seq)have been deposited in Dryad (https://doi.org/10.5061/dryad.3j9kd51t7).

The following dataset was generated:

| Author(s) | Year | Dataset title | Dataset URL | Database and Identifier |
|---|---|---|---|---|
| Samarjit J, Mainak M, Sagar M, Bhavana G, Kaval Reddy P, Lekha K, Neha J, Abhishek C, Vani S, Chandrasekhar K, Kumaravel S | 2024 | PITAR, a DNA damage-inducible Cancer/Testis long noncoding RNA, inactivates p53 by binding and stabilizing TRIM28 mRNA | https://doi.org/10.5061/dryad.3j9kd51t7 | Dryad Digital Repository, 10.5061/dryad.3j9kd51t7 |

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
