## [Editor Report · eLife assessment]

This **important** study reports, with **convincing** evidence, that a long non-coding RNA disrupts the activity of the tumor suppressor p53 to contribute to the growth and therapeutic response of glioblastoma. The work will be relevant to scientists working on non-coding RNAs and brain tumors.

---

## [Referee Report · Reviewer #1 (Public review)]

The authors characterized a new non-coding RNA, which they named as PITAR. They first showed that the PITAR expression levels are higher in glioblastoma, and then demonstrated that knockdown of PITAR in glioblastoma cells decreased cell growth, induced G0/G1 arrest and apoptosis. They further identified the E3 ubiquitin ligase TRIM28 is the target of PITAR, and showed that PITAR bound to the TRIM28 mRNA and regulated the stability and expression of the latter. Since TRIM28 has been reported to be an E3 ubiquitin ligase for the tumor suppressor p53, the authors tried to link the PITAR function to p53 regulation. They showed that one PITAR siRNA increased the levels of p53 and p21, and the stability of p53, and these effects could be diminished by overexpression of TRIM28. They also showed that PITAR overexpression decreased the levels of adriamycin-induced p53/p21 expression and reversed DNA damage-induced G2/M arrest. Lastly, the authors showed that PITAR siRNA decreased the growth of glioblastoma, while PITAR overexpression increased glioblastoma growth and counteracted temozolomide for its anti-glioblastoma activity.

Overall, the manuscript has provided evidence supporting the important role of PITAR in the regulation of the growth of glioblastoma. The results supporting the regulation of PITAR on TRIM28 appear to be convincing. However, some weaknesses are also noted.

(1) More than one siRNA/shRNA should be used in critical experiments. For example, Fig 7A-E are important experiments demonstrating PITAR suppresses tumor growth. It is compelling that the siPITAR tumors disappeared at the end of the experiment. While this might be due to apoptosis, using another siRNA to confirm the results would be necessary. The authors may also need to use this model to test their hypothesis that PITAR regulates tumor growth through p53. They can check p53, p21, apoptosis levels in tumor sections.

(2) The data supporting that PITAR downregulates p53 stability and activity can be strengthened. The half-life of endogenous p53 protein is generally 20-30 min, and thus the cycloheximide chase experiments (Fig 5E) need to use shorter treatment time. The ubiquitinated p53 bands are not clear (Fig 5F), and the data suggesting that PITAR regulates p53 ubiquitination are not convincing. While the p53 protein level was largely altered by PITAR/TRIM28, the mRNA levels of its target genes, including p21 and MDM2 only marginally changed (Fig S6D). Other p53 targets, particularly proapoptotic genes, may need to be examined.

(3) The model depicting the role of PITAR in the cellular response to DNA-damaging agents is confusing. If DNA damaging agents like TMZ induce PITAR to inactivate p53, PITAR overexpression would confer TMZ resistance. However, Fig 7G did not support this. While the experimental design is quite problematic given that U87 cells already express a high level of PITAR, PITAR-overexpressing cells were still sensitive to TMZ treatment (this is apparent when checking the images in Fig 7F, although the large error bars shown in Fig 7G may lead to a "not significant" conclusion). The authors may need to test whether PITAR downregulation, which would increase p53 activity, has any effects on TMZ-insensitive tumors. Such results are more therapeutically relevant. It would also be helpful if the authors test whether PITAR is overexpressed in TMZ-resistant clinical samples.

---

## [Referee Report · Reviewer #2 (Public Review)]

This study established an alternate way of p53 inactivation and proposed PITAR as a potential therapeutic target, so the impact is high. In addition, this manuscript has apparent strengths, including a logically designed research strategy, in vitro and in vivo study, and well-designed control.

This manuscript identified a long noncoding RNA, PITAR (p53 Inactivating TRIM28 associated RNA), as an inhibitor of p53. PITAR is highly expressed in glioblastoma (GBM) and glioma stem-like cells (GSC). The authors found that TRIM28 mRNA, which encodes a p53-specific E3 ubiquitin ligase, is a direct target of PITAR. PITAR interaction with TRIM28 RNA stabilized TRIM28 mRNA, which resulted in increased TRIM28 protein levels, enhanced p53 ubiquitination, and attenuated DNA damage response. While PITAR silencing inhibited the growth of WT p53 containing GSCs in vitro and reduced glioma tumor growth in vivo, its overexpression enhanced the tumor growth and promoted resistance to Temozolomide. DNA damage also activated PITAR, in addition to p53, thus creating an incoherent feedforward loop. Together, this study established an alternate way of p53 inactivation and proposed PITAR as a potential therapeutic target.

P53 is a well-established tumor suppressor gene contributing to cancer progression in many human cancers. It plays a vital role in preserving genome integrity and inhibiting malignant transformation. p53 is mutated in more than 50% of human cancers. In cancers that do not carry mutations in p53, the inactivation occurs through other genetic or epigenetic alterations. Therefore, further study of the mechanism of regulation of wt-p53 remains vital in cancer research. This study identified a novel LncRNA PITAR, which is highly expressed in glioblastoma (GBM) and glioma stem-like cells (GSCs) and interacts with and stabilizes TRIM28 mRNA, which encodes a p53-specific E3 ubiquitin ligase. TRIM28 can inhibit p53 through HDAC1-mediated deacetylation and direct ubiquitination in an MDM2-dependent manner. Thus, the overall impact of this study is high because of the identification of a novel mechanism in regulating wt-p53.

The other significant strengths of this manuscript included an apparent research strategy design and a clearly outlined and logically organized research approach. They provided both the in vitro and in vivo studies to evaluate the effect of PITAR. They offered reasonable control of the study by validating the results in cells with mutant p53. They also performed a rescue experiment to confirm the PITAR and TRIM28 relationship regulating p53. The conclusions were all supported by solid results. The overall data presentation is clear and convincing.

---

## [Author Response]

The following is the authors’ response to the original reviews.

**Reviewer #1 (Public Review):**
(1) Only one PITAR siRNA was tested in majority of the experiments, which compromises the validity of the results.

We thank the reviewer for this comment. We have now used two siRNAs to demonstrate PITAR functions in various assays. In the revised manuscript, we carried out additional experiments with two siRNAs, and the results are presented in Figures 2C, D, F, G, H, I, and J; Figures 5A, B, Supplementary Figure 2B, C, D, E, and F.

(2) Some results are inconsistent. For example, Fig 2G indicates that PITAR siRNA caused G1 arrest. However, PITAR overexpression in the same cell line did not show any effect on cell cycle progression in Fig 5I.

The reason for the fact that PITAR silencing showed a robust G1 arrest, unlike PITAR overexpression, is as follows. Since glioma cells overexpress PITAR (which keeps the p53 suppressed), silencing PITAR (which will elevate p53 levels) in glioma cells shows a robust phenotype in cell cycle profile (in the form of increased G1 arrest). In contrast, the overexpression of PITAR in glioma cells fails to show robust changes in the cell cycle profile because glioma cells already have high levels of PITAR.

(3) The conclusion that PITAR inactivates p53 through regulating TRIM28, which is highlighted in the title of the manuscript, is not supported by convincing results. Although the authors showed that a PITAR siRNA increased while PITAR overexpression decreased p53 level, the siRNA only marginally increased the stability of p53 (Fig 5E). The p53 ubiquitination level was barely affected by PITAR overexpression in Fig 5F.

We disagree with the fact that PITAR silencing only marginally increased the stability of p53. In the cycloheximide experiment in Figure 5E, the half-life of p53 is increased by 60 % (50 mins to 120 mins), which is quite significant in altering the DNA damage response by p53. Further, we also want to point out that the other arm of p53 degradation by Mdm2 remains intact under these conditions. We also provide an improved p53 ubiquitination western blot in the revised version (Figure 5F).

(4) To convincingly demonstrate that PITAR regulates p53 through TRIM28, the authors need to show that this regulation is impaired/compromised in TRIM28-knockout conditions. The authors only showed that TRIM28 overexpression suppressed PITAR siRNA-induced increase of p53, which is not sufficient.

We thank the reviewer. In the revised manuscript, we demonstrate that PITAR overexpression fails to inhibit p53 in TRIM28 silenced cells (Supplementary Figure 5G; Figure 5K, L, M, N).

(5) Note that only one cell line was investigated in Fig 5.

In revised manuscript, the impact of PITAR silencing and PITAR overexpression on p53 functions are demontsrared for one more glioma cell line (Supplemenatry Figure 5B, C, D, and E).

(6) Another major weakness of this manuscript is that the authors did not provide any evidence indicating that the glioblastoma-promoting activities of PITAR were mediated by its regulation of p53 or TRIM28 (Fig 6 and Fig 7). Thus, the regulation of glioblastoma growth and the regulation of TRIM28/p53 appear to be disconnected.

We would like to respectfully disagree with the reviewer on this particular point. We have indeed provided the following evidence in the first version of the manuscript: glioblastoma-promoting activities of PITAR were mediated by its regulation of p53 or TRIM28.

(1) To show the importance of p53:

We show that PITAR silencing failed to inhibit the colony growth of p53-silenced U87 glioma cells (U87/shp53#1). We also show that while PITAR silencing decreased TRIM28 RNA levels in U87/shNT and U87/shp53#1 glioma cells, it failed to increase CDKN1A and MDM2 (p53 targets) at the RNA level in U87/shp53#1 cells unlike in U87/siNT cells (Supplementary Figure 6 Panels A, B, C, and D).

(2) To show the importance of TRIM28 and p53:

The importance of p53 is also demonstrated in the context of patient-derived GSC lines. We demonstrate that PITAR silencing-induced reduction in the neurosphere growth (WT p53 containing patient-derived GSC line) is accompanied by a reduction in TRIM28 RNA and an increase in the CDKN1A RNA without a change in p53 RNA levels (Supplementary Figure 7 Panels A, B, C, D, and E). We also demonstrate that PITAR overexpression-induced neurosphere growth is accompanied by an increase in the TRIM28 RNA, and a decrease in CDKN1A RNA without a change in p53 RNA levels (Supplementary Figure 7 Panels F, G, H, and I). However, PITAR silencing failed to decrease neurosphere growth in mutant p53 containing GSC line (MGG8) (Supplementary Figure 7 Panels J, K, L, M, N, and F).

(3) We show that the TRIM28 protein level is drastically reduced in small tumors formed by U87/siPITAR cells (Supplementary Figure 7 Panel E).

(4) We show that glioma tumors formed by U87/PITAR OE cells express high levels of TRIM28 protein but reduced levels of p21 protein (Supplementary Figure 7 Panel B).

Further, we did additional experiments to prove the importance of TRIM28.

In the revised manuscript, we have carried out an additional experiment to prove the requirement of TRIM28 for tumor-promoting functions of PITAR overexpression. Earlier, we have shown that exogenous overexpression of PITAR promotes glioma tumor growth and imparts resistance to Temozolomide chemotherapy (Figure 7F and G; Supplementary Figure 9A and B). In the revised manuscript, we show that the tumor growth-promoting function of PITAR overexpression requires TRIM28. U87-Luc/PITAR OE cells formed a larger tumor compared to U87-Luc/VC cells (Figure 7H, and I; compare red line with blue line). U87-Luc/shTRIM28 cells formed very small-sized tumors (Figure 7H, and I; compare green line with blue line). Further, PITAR overexpression (U87-Luc/PITAR OE) was less efficient in promoting glioma tumor growth in TRIM28 silenced cells (Figure 7H, and I; compare pink line with red line). Thus, we prove that, as a whole, TRIM28 mediates the tumor growth-promoting functions of PITAR.

(7) It is not clear what kind of message the authors tried to deliver in Fig 7F/G. Based on the authors' hypothesis, DNA-damaging agents like TMZ would induce PITAR to inactivate p53, which would compromise TMZ's anti-cancer activity. However, the data show that TMZ was very effective in the inhibition of U87 growth. The authors may need to test whether PITAR downregulation, which would increase p53 activity, have any effects on TMZ-insensitive tumors. Such results are more therapeutically relevant.

Reviewer #1 rightly pointed out that TMZ induces PITAR expression, which should compromise TMZ's anti-cancer activity.

We demonstrate the same as below:

Figure 7F&G demonstrates the following two facts:1. PITAR overexpression increases the glioma-tumor growth (Figure 7G, compare red line with the blue line), 2. PITAR overexpressing glioma tumors are resistant to TMZ chemotherapy (Figure 7G, compare the pink line with the green line).

In addition, Figure 7 F and G also demonstrate that TMZ treatment of tumors formed by U87/VC glioma cells inhibited the growth but not eliminated the tumor growth completely (compare pink line with blue line). We believe that the inability of TMZ to eliminate the tumor growth completely is because of the chemoresistance imparted by the DNA damage induced PITAR.

Further, in Figure 2I, we indeed show that PITAR-silenced cells are more sensitive to TMZ and Adriamycin chemotherapy.

(8) Lastly, the model presented in Fig 7H is confusing. It is not clear what the exact role of PITAR in the DNA damage response based on this model. If DNA damage would induce PITAR expression, this would lead to inactivation of p53 as revealed by this manuscript. However, DNA damage is known to activate p53. Do the authors want to imply that PITAR induction by DNA damage would help to bring down the p53 level at the end of DNA damage response? The presented data do not support this role unfortunately.

We respect the views and questions raised by the reviewer.

We would like explain as below the importance of our model.

Yes, it is true that DNA damage induces p53. We show here that DNA damage also induces PITAR in a p53-independent manner, which, in turn, inhibits p53. Here is our explanation. Even though DNA damage activates p53, there exists an autoregulatory negative feedback loop that controls the extent and duration of p53 response to DNA damage (Wu et al., 1993; Haupt et al., 1997; Kubbutat, Jones and Vousden, 1997; Zhang et al., 2009). It is proposed that the p53-Mdm2 feedback loop generates a “digital clock” that releases well-timed quanta of p53 until the damage is repaired or the cell dies (Lahave et al., 2004). In addition, it has also been shown that TRIM28, through its association with Mdm2, also contributes to p53 inactivation (Wang et al., 2005b; Czerwińska, Mazurek, and Wiznerowicz, 2017).

Based on the above reports and our current work, we propose that DNA damage-induced PITAR, through its ability to increase the TRIM28 levels, contributes to the control of the DNA damage response of p53 along with Mdm-2. The difference is as follows: Since Mdm-2 is also a transcriptional target of p53, the p53-Mdm-2 axis is an autoregulatory negative feedback loop to control the DNA damage response by p53. In contrast, PITAR is not a transcriptional target of p53, and DNA damage-induced activation of PITAR is p53-independent. Hence, the PITAR-TRIM28 axis in controlling the DNA damage response of p53 creates an Incoherent feedforward regulatory network. The experimental evidence provided in the revised manuscript is as follows: (1) We have already (the first version of the manuscript) shown that exogenous overexpression of PITAR significantly inhibits DNA damage-induced p53 (Figures 6A, B, C, and D). (2) In the revised manuscript, we show that the DNA damage response of p53 (duration and extent of p53 activation after a pulse of ionizing radiation) in PITAR-silenced cells follows similar kinetics in terms of duration, but the extent of p53 activation was much stronger (Supplementary figures 8H, I, J, and K). This is because the TRIM28 component in TRIM28/Mdm-2 axis is compromised as PITAR silencing reduces the TRIM28 levels. (3) We also demonstrate that DNA damage-induced TRIM28 is dependent on PITAR (Figure 6K; Supplementary Figure 5G)

Reviewer #1(Recommendations For The Authors):(1) Fig 7A, what is the explanation for the observation that tumors disappeared in most of the mice in the siPITAR group? Did the authors check if apoptosis was induced here?

We agree to the point that the lack of tumor growth in the siPITAR group is likely due to the induction of apoptosis. We would like to point out that in vitro experiments indeed demonstrate that PITAR silencing induces apoptosis in Figure 2H and Supplementary Figure 2F.

(2) The authors need to explain why Fig 6 used a cell line different from other experiments. It would be better to check other cell lines.

The purpose of RG5 and MGG8 is as follows. (1) We wanted to establish the growth-promoting functions of PITAR in patient-derived GSC lines. (2) We also wanted to show the importance of WT p53 for the growth-promoting functions of PITAR.

However, in the revised manuscript we moved this portion under the subsection “PITAR inhibits p53 protein levels by its association with TRIM28 mRNA“.

Further,the experiments related to DNA damage induced activation of PITAR in p53-independent manner and its impact on DNA damage response by p53 is moved to a new section entitled “PITAR is induced by DNA damage in a p53-independent manner, which in turn diminishes the DNA damage response by p53”

(3) It would be more convincing if the authors could test more p53 target genes in addition to p21.

We thank the reviewer for this comment and the specific suggestions for checking additional p53 targets. In the revised manuscript, we have checked the MDM2 transcript levels in Supplementary Figure 6D.

Reviewer #2 (Recommendations For The Authors):(1) In the text, they mentioned " Figure 4J". There is no Figure 4J in Figure 4. It may be Figure 4K.

We thank reviewer #2. We corrected this information in the revised manuscript.

(2) The molecular weight markers in Western blots were missed in several Figure panels, including Figure 4J, Figure 5K, and Supple. Figure 3B, Supple. Figure 5G, H, Supple. Figures 6A and 7A.

We thank reviewer #2, and we have included the molecular weight markers in all the mentioned figures.